# Highly efficient photothermal nanoagent achieved by harvesting energy via excited-state intramolecular motion within nanoparticles

Zheng Zhao[1], Chao Chen[2], Wenting Wu[3], Fenfen Wang[4], Lili Du [5,6], Xiaoyan Zhang[2], Yu Xiong[7], Xuewen He[1], Yuanjing Cai [1], Ryan T.K. Kwok[1], Jacky W.Y. Lam[1], Xike Gao[3], Pingchuan Sun [4], David Lee Phillips[5], Dan Ding[2] & Ben Zhong Tang [1,7]

The exciting applications of molecular motion are still limited and are in urgent pursuit, although some fascinating concepts such as molecular motors and molecular machines have been proposed for years. Utilizing molecular motion in a nanoplatform for practical application has been scarcely explored due to some unconquered challenges such as how to achieve effective molecular motion in the aggregate state within nanoparticles. Here, we introduce a class of near infrared-absorbing organic molecules with intramolecular motion-induced photothermy inside nanoparticles, which enables most absorbed light energy to dissipate as heat. Such a property makes the nanoparticles a superior photoacoustic imaging agent compared to widely used methylene blue and semiconducting polymer nanoparticles and allow them for high-contrast photoacoustic imaging of tumours in live mice. This study not only provides a strategy for developing advanced photothermal/photoacoustic imaging nanoagents, but also enables molecular motion in a nanoplatform to find a way for practical application.

[1] Department of Chemistry, The Hong Kong Branch of Chinese National Engineering Research Center for Tissue Restoration and Reconstruction, Institute of Molecular Functional Materials, Division of Life Science and State Key Laboratory of Molecular Neuroscience, The Hong Kong University of Science and Technology, Clear Water Bay, Kowloon, Hong Kong 999077, China. [2] State Key Laboratory of Medicinal Chemical Biology, Key Laboratory of Bioactive Materials, Ministry of Education, and College of Life Sciences, Nankai University, Tianjin 300071, China. [3] Key Laboratory of Synthetic and Self-assembly Chemistry for Organic Functional Molecules, Shanghai Institute of Organic Chemistry, Chinese Academy of Science, 345 Lingling Road, Shanghai 200032, China. [4] Key Laboratory of Functional Polymer Materials, Ministry of Education, College of Chemistry, Nankai University, Tianjin 300071, China. [5] Department of Chemistry, The University of Hong Kong, Pokfulam Road, Hong Kong 000000, China. [6] Institute of Life Sciences, Jiangsu University, Zhenjiang 212013, China. [7] Guangdong Provincial Key Laboratory of Brain Science, Disease and Drug Development, Shenzhen Research Institute, No. 9 Yuexing 1st RD, South Area, Hi-tech Park, Nanshan, Shenzhen 518057, China. These authors contributed equally: Zheng Zhao, Chao Chen, Wenting Wu. Correspondence and requests for materials should be addressed to D.D. (email: dingd@nankai.edu.cn) or to B.Z.T. (email: tangbenz@ust.hk)

The ability to transform low density light energy source to heat makes photothermal materials promising candidates for many important applications such as seawater desalination, photothermal–electrical and photothermal–mechanical converters, photoacoustic (PA) imaging, and photothermal therapy[1–5]. In the biomedical field, PA imaging that detects the photothermally generated ultrasound signal has recently received considerable attention as it surpasses the optical diffusion limit, and permits disease diagnosis in deeper tissue with higher spatial resolution[6–15]. The PA effect of the materials that originated from heat generation is closely associated with several parameters of the contrast agents including the absorption coefficient, Grüneisen coefficient, and the non-radiative decay efficiency according to the PA equation (see Supplementary Equation 1 and 2)[16]. By either increasing the absorption coefficient or tuning Grüneisen coefficient, scientists have successfully improved the performance of the PA imaging agents[17–19]. For examples, Pu and Rochford et al. found that the increase of the ground state and excited absorption coefficient, respectively, could obviously enhance the PA intensity[17,18]. Aoki and co-workers demonstrated that the thermal confinement in the nanoparticle and the large Grüneisen parameter of the material of nanosized PA contrast agents helped improve their PA performance significantly[19]. Besides, another important parameter is the non-radiative decay efficiency of the PA contrast agent, which determines how much absorbed light energy can be converted to heat[16].

Among various PA imaging agents, great interest has been placed on nanoparticles (NPs) based on organic π-conjugated molecules or polymers rather than their inorganic counterparts due to their excellent biocompatibility, easily tunable band gap, and accessible structural–property relationship[5,16,17]. Some popularly used molecules, such as indocyanine green (ICG) and methylene blue (MB), have been approved by the Food and Drug Administration for clinical use[20,21]. However, these planar molecules generally show bright emission in solution. Even in the aggregate state, they may still emit intensely only when strong face-to-face π–π stacking such as H-aggregation occurs, the molecules exhibit efficient non-radiative decay according to Kasha's exciton model[22]. Unfortunately, the formation of such intermolecular interaction is hard to control, and in most cases, the aggregates formed in organic NPs are randomly aligned and amorphous, to show both insufficient radiative decay and non-radiative decay[23]. Consequently, they are not ideal agents for either PA imaging or fluorescence imaging. From the perspective of practical/clinical application, new molecular design concept to advanced organic NPs with efficient non-radiative decay to generate PA signal is highly desirable.

Exploring the motion of matters at the molecular level has been pursued by scientists for years since it correlates with many basic physical and chemical characteristics of matters[24–27]. The early dispute on the intrinsic nature of Brownian motion has indirectly confirmed the concrete existence of atoms and molecules in materials[24], while the latest breakthrough in molecular machine possibly bring us another giant leap in academics although exciting applications are still under investigation[28–31]. Recently, the study of excited-state intramolecular motion such as rotation and vibration has attracted increasing attention because of their significant role in developing sophisticated functional materials such as molecular rotors, stimuli-responsive polymers, aggregation-induced emission luminogens (AIEgens), and other smart fluorescent materials[32–35]. These ideas generally utilize the conversion of photo energy to mechanical energy or luminescence to achieve a specific function. Nevertheless, active intramolecular motion theoretically also promotes efficient non-radiative decay to release the excitation energy as heat[36], which potentially enables the development of photothermal materials

and provides an outlet for the application of molecular motion. However, so far, this kind of possibility has been scarcely explored. On the other hand, although NPs hold the momentous merit that preferentially accumulate in tumour tissues in vivo via the enhanced permeability and retention (EPR) effect[11], how to realize molecules with effective excited-state intramolecular motion in the aggregate state within NPs is greatly challenging. Unfortunately, currently available molecular rotors and vibrators generally show active intramolecular motion in solution, which is significantly suppressed in the aggregate and solid states[37]. This thus motivates us to develop an approach to alternative NPs with highly boosted photothermy by virtue of internally efficient excited-state intramolecular motion.

Here, we report a molecular guideline to develop organic photothermal nanoagents with highly efficient photothermal conversion and PA effect by taking advantage of active intramolecular motion in the aggregate state within NPs. Two near infrared (NIR)-absorbing organic molecules with donor–acceptor structures, namely 2TPE-NDTA and 2TPE-2NDTA, have been rationally designed. These molecules can undergo highly effective intramolecular motion in the solid state and aggregate state within NPs, which leads to the loss of most absorbed energy harvested as heat production through the boosted non-radiative decay process. As a result, the 2TPE-2NDTA-doped NPs exhibit a higher ability in photothermal conversion and generating PA signal than several well-established, high-performing PA imaging agents including semiconducting polymer NPs and MB. In vivo study verifies that 2TPE-2NDTA-doped NPs give excellent performance in visualizing tumours in a high-contrast manner via PA imaging. This study not only demonstrates that intramolecular motion can be used for developing superior photothermal/photoacoustic imaging nanoagents, but also finds a way to the practical application of molecular motion in a nanoplatform.

## Results

**Molecular design and synthesis**. The molecular design of 2TPE-NDTA and 2TPE-2NDTA is shown in Fig. 1. Tetraphenylethylene (TPE) is selected as it undergoes active excited-state intramolecular motion. The acceptors are naphthalene diimide-fused 2-(1,3-dithiol-2-ylidene)acetonitriles (NDTA and 2NDTA) with long alkyl chains[38,39]. The large π-conjugation and strong electron-withdrawing ability of the acceptors when combining with that of TPE will contribute long wavelength absorption, high molar absorptivity and strong twisted intramolecular charge transfer (TICT)[32] effect to the resulting molecules[40–43]. The long alkyl chain is expected to enable the intermolecular spatial isolation of the molecules in the aggregate state to produce some necessary rooms to promote free intramolecular motion[37,44,45].

The synthetic routes to 2TPE-NDTA, 2TPE-2NDTA, and other involved molecules are presented in Supplementary Fig. 1 and 2. Suzuki–Miyaura coupling reaction between dibrominated NDTA/2NDTA and the boric acid of TPE produced the target compounds (2TPE-NDTA and 2TPE-2NDTA) in yields of 77% and 44%, respectively. 2NDTA was synthesized through homocoupling of monobromo-substituted NDTA in a yield of 70%[39]. In the presence of bromine and chloroform at room temperature, NDTA was efficiently brominated, to give the mono-/dibromo-substituted NDTA in yields of 42% and 45%, respectively. The chemical structures of 2TPE-NDTA and 2TPE-2NDTA are shown in Fig. 1a. These molecules were fully characterized by NMR spectroscopy and high-resolution mass spectrometry and melting point measurement with satisfactory results (Supplementary Fig. 3–19). Comparing with the precursor (NDTA), both 2TPE-NDTA and 2TPE-2NDTA exhibit improved solubility in

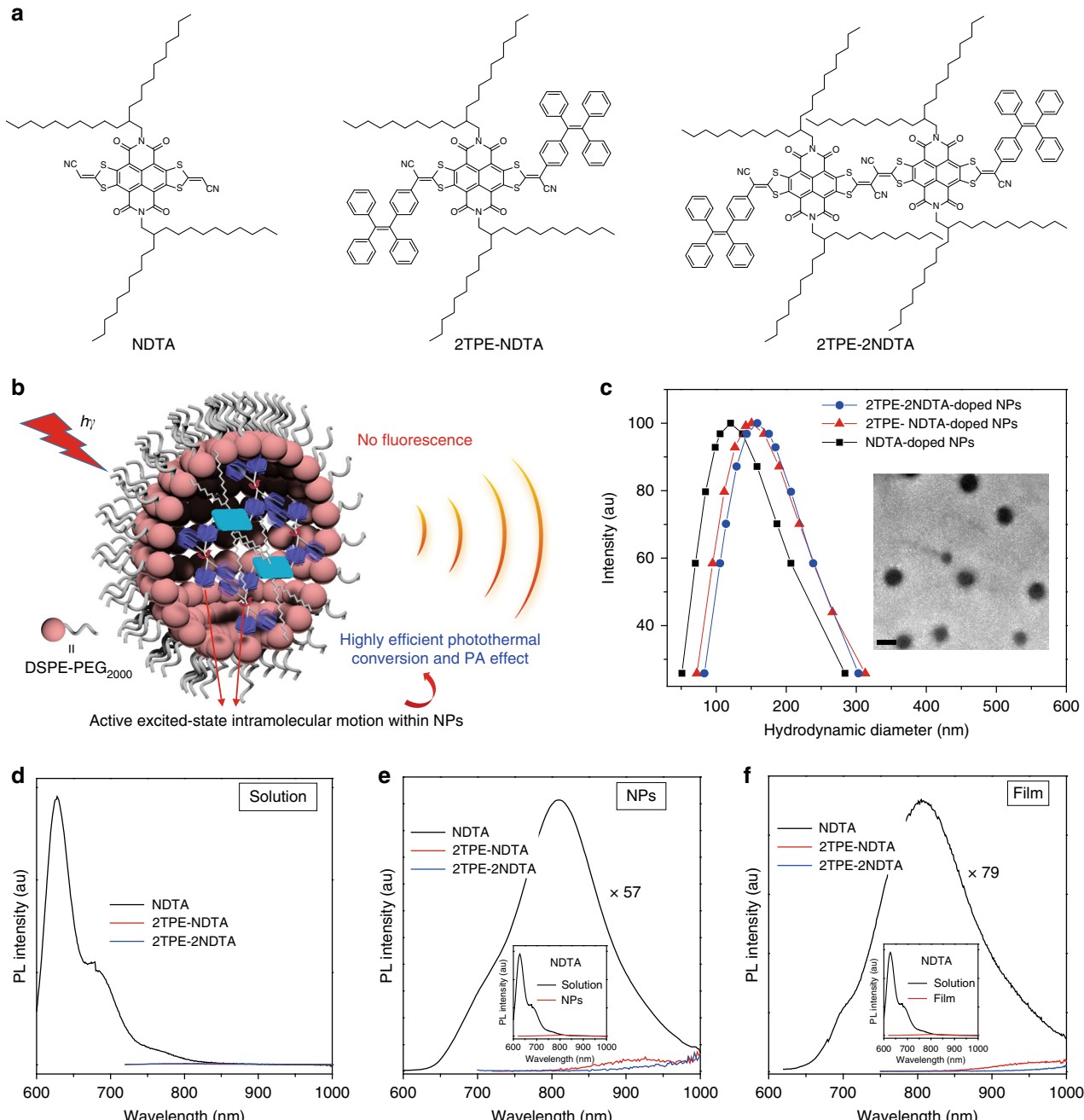

**Fig. 1** Preparation and characterization of the compounds and NPs. **a** Molecular structures of NDTA, 2TPE-NDTA, and 2TPE-2NDTA. **b** Schematic of 2TPE-NDTA-doped NP, showing active excited-state intramolecular motion within NPs. **c** Dynamic light scattering profiles of the NPs indicated. Inset: transmission electron microscopy image of 2TPE-2NDTA-doped NPs. Scale bar, 200 nm. **d–f** The photoluminescence (PL) spectra of NDTA (black), 2TPE-NDTA (red), and 2TPE-2NDTA (blue) in THF solution (**d**), the doped NPs (**e**) in water and thin films (**f**). Inset: the contrast of the PL intensity of the as-prepared NPs in water and in a THF solution of NDTA (**e**), and the contrast of the PL intensity of thin film and THF solution of NDTA (**f**)

common organic solvents such as tetrahydrofuran (THF) and chloroform and better processability. This demonstrates their relatively weaker intermolecular interaction.

**Photophysical properties**. The optical properties of 2TPE-NDTA and 2TPE-2NDTA were characterized by photoluminescence (PL) and UV–vis–NIR spectroscopies and compared with NDTA. To evaluate their optical characteristics in the aggregate state, nano-precipitation strategy was employed to formulate these molecules into water-soluble NPs using 1,2-distearoyl-sn-glycero-3-phosphoethanolamine-N-[methoxy-(polyethylene glycol)-2000]

(DSPE-PEG$_{2000}$) as the encapsulation matrix (Fig. 1b)[46]. The afforded NDTA-doped, 2TPE-NDTA-doped, and 2TPE-2NDTA-doped NPs possess an average diameter of about 125, 152, and 156 nm, respectively (Fig. 1c). As shown in Fig. 1d–f, NDTA emits brightly at around 630 nm in dilute THF solution, and its PL quantum yield (QY) was determined as $\Phi_{soln, NDTA} = 17\%$ using a calibrated integrating sphere. While in the aggregate state, both its NPs and film show much decreased PL intensity but largely red-shifted emission peaks at around 810 nm ($\Phi_{NPs, NDTA} = 2.5\%$, $\Phi_{solid, NDTA} = 4\%$), which suggests the formation of *J*-aggregation with aggregation-caused quenching (ACQ) characteristics

(Supplementary Fig. 20)[22]. This is reasonable since the large π-conjugation and planar structure of NDTA is favorable for strong π–π stacking. However, after coupling with TPE, both the dilute THF solution and aggregates (NPs and thin film) of 2TPE-NDTA ($\Phi_{soln, 2TPE-NDTA} = 0.4\%$, $\Phi_{NPs, 2TPE-NDTA} = 0.4\%$, $\Phi_{solid, 2TPE-NDTA} = 1\%$) and 2TPE-2NDTA ($\Phi_{soln, 2TPE-2NDTA} = 0.4\%$, $\Phi_{NPs, 2TPE-2NDTA} = 0.3\%$, $\Phi_{solid, 2TPE-2NDTA} = 0.9\%$) emit almost no light (Fig. 1d–f), indicating that the non-radiative decay is dominated for the exciton relaxation of 2TPE-NDTA and 2TPE-2NDTA even in the solid state. This is quite different from the typical design of AIE molecules, where the introduction of TPE usually transform the ACQ molecules to AIE ones such as the TPE-substituted naphthalene diimide by Bhosale and co-workers[40,41].

Additionally, the molar extinction coefficients of 2TPE-NDTA ($50,600\ mol^{-1}\ L\ cm^{-1}$) and 2TPE-2NDTA ($67,800\ mol^{-1}\ L\ cm^{-1}$) were comparable to that of NDTA ($67,822\ mol^{-1}\ L\ cm^{-1}$) (Supplementary Fig. 21), revealing their excellent light-harvesting ability, which could benefit to their PA signal according to the PA equation[16]. From solution to the aggregates, the absorption profile and absorption maximum of NDTA exhibit large red-shifts to support the formation of *J*-aggregates (Supplementary Fig. 22). Unlike NDTA, the absorption profile of 2TPE-NDTA is broadened but the absorption maximum changes little from solution to aggregate state, indicating that the introduction of TPE hinders the strong π–π stacking of 2TPE-NDTA molecules. The absorption spectrum of 2TPE-2NDTA in the solution state is almost identical to that of 2TPE-NDTA, manifesting that 2TPE-2NDTA has comparable conjugation as 2TPE-NDTA in the solution state. This could be attributed to the twisted structure of 2NDTA, which hampers the conjugation of the whole molecule[47]. Upon aggregation, the molecular planarity of 2TPE-2NDTA is significantly improved to result in a large red-shift of the absorption spectrum (Supplementary Fig. 22). It is noteworthy that the absorption of 2TPE-2NDTA-doped NPs can be well extended to 880 nm, with a strong absorption at 808 nm, which is well matched with the excitation wavelength of commercial NIR laser source. Such strong and long-wavelength absorption are benefitted to PA imaging and photothermal applications.

**Excited-state dynamics of the molecules**. To better understand the underlying mechanism of the non-emissive behaviors of 2TPE-NDTA and 2TPE-2NDTA, femtosecond time-resolved fluorescence (fs-TRF) and femtosecond transient absorption (fs-TA) experiments were implemented to investigate their excited-state relaxation behavior[48,49]. The fs-TRF spectrum of NDTA in THF with a time delay of 565 fs after 400 nm excitation exhibits a broad emission band centered at 630 nm (Fig. 2a–c), whose intensity increases with increasing the delay time to up to 19 ps due to the relaxation of the $S_n$ excited state to the $S_1$ state. Further gradually lengthening delay time to up to 5.9 ns, the emission band becomes weaker stemmed by the delay of $S_1$. The kinetics at 630 nm can be fitted satisfactorily by a two-exponential function with time constants of 4.1 ps and 1.5 ns, respectively. Therefore, the lifetime of $S_1$ of NDTA is around 1.5 ns. The fs-TA results of NDTA agree well with that of fs-TRF as shown in Supplementary Fig. 23. A similar transition from $S_n$ to $S_1$ with a timescale of 687 fs to 8.09 ps was observed in 2TPE-NDTA (Fig. 2d–f). The band at around 500 nm decreases with a slight red-shift while the one at around 750 nm increases dramatically with a small blue-shift, suggesting a rapid conformation change. However, unlike NDTA, the transient absorption bands of $S_1$ of 2TPE-NDTA (500 and 750 nm) decay at a much faster rate. And global analysis of the decay kinetics at all wavelengths indicates that a

satisfactory fitting requires two exponential functions to give time constants of 2.6 ps ($\tau_1$) and 53.8 ps ($\tau_2$). The first time constant (2.6 ps) originates from the internal conversion (IC) from $S_n$ to $S_1$, while the second one (53.8 ps) is assigned to the IC process from $S_1$ to $S_0$. Compared with NDTA with a long-lived (1.5 ns) singlet excited states, the $S_1$ of 2TPE-NDTA favors to undergo radiationless process. This suggests that the TPE units accelerate the IC process by rotation its phenyl rings. As displayed in Fig. 2g–i, 2TPE-2NDTA behaves similar to 2TPE-NDTA, and shows absorption bands mainly at 500 and 700 nm. The global analysis of the kinetics at all wavelengths shows that a satisfactory fitting requires two exponential functions to give short time constants of 2.0 ps ($\tau_1$) and 42.1 ps ($\tau_2$) due to the rapid IC from $S_n$ to $S_1$ and $S_1$ to $S_0$, respectively.

To further demonstrate the role of intramolecular motion in promoting the non-radiative decay of 2TPE-NDTA and 2TPE-2NDTA, we investigated their emission behaviors in dilute THF at 77 K, at which their intramolecular motion will be strongly suppressed (Supplementary Fig. 24). The results indicate that both 2TPE-NDTA and 2TPE-2NDTA show intense fluorescence at around 800 nm at 77 K. Moreover, 2TPE-NDTA and 2TPE-2NDTA exhibit no light in polar solvents but enhanced emission in non-polar solvents. This is because non-polar solvents destabilize their charge separation state to restrict their conformation change to polarized conformation[50]. This causes strong suppression on the intramolecular rotation of 2TPE-NDTA and 2TPE-2NDTA to lead to their enhanced fluorescence (Supplementary Fig. 25). These results imply that the intramolecular motion plays a key role in the non-radiative decay of 2TPE-NDTA and 2TPE-2NDTA.

**Theoretical calculation**. The electronic structures in the ground state ($S_0$) at the level of B3LYP/6-31G(d,p) were investigated to decipher the different optical properties of the studied molecules (Supplementary Fig. 26). NDTA exhibits a planar structure. Its highest occupied molecular orbitals (HOMO) and lowest unoccupied molecular orbitals (LUMO) are contributed by orbitals from the whole molecules and overlap in a large extent to result in the emissive nature of its solution. 2TPE-NDTA, on the other hand, shows a dumbbell shape molecular conformation owing to the twisted structure of TPE, which hampers the occurrence of strong intermolecular π–π stacking. In addition, the orbitals of the HOMO of 2TPE-NDTA are mainly located on the TPE moieties whereas the LUMO are mainly contributed by orbitals of the NDTA moiety. This reveals an inherent ICT effect. 2TPE-2NDTA shows an even more twisted conformation. Besides the twisted TPE moieties, the central acceptor unit of 2NDTA is also highly twisted with a dihedral angle of 120°. The HOMO orbitals are located on one of the TPE moieties, while the LUMO orbitals are contributed mainly by the 2NDTA moiety. This also suggests an ICT effect. The twisted structure and the ICT effect of 2TPE-NDTA and 2TPE-2NDTA are beneficial to active intramolecular motion since they not only hinder the intermolecular π–π interaction to facilitate the spatial isolation of molecules but also enhance the TICT effect to finally contribute to the non-radiative relaxation of the excitons[50].

**Molecular motion behavior of the aggregates**. Solid-state nuclear magnetic resonance (SSNMR) was employed to study the intramolecular motion of 2TPE-NDTA and 2TPE-2NDTA in the solid state. Since both 2TPE-NDTA and 2TPE-2NDTA possess no aromatic hydrogen atoms except those of TPE, it is easy to identify the $^{13}C$ signals of the TPE moieties by $^{13}C$ CPMAS NMR spectroscopy and probe their intramolecular motion[51]. As shown in Fig. 3a, the $^{13}C$ CPMAS NMR spectra at 5 kHz with toss and

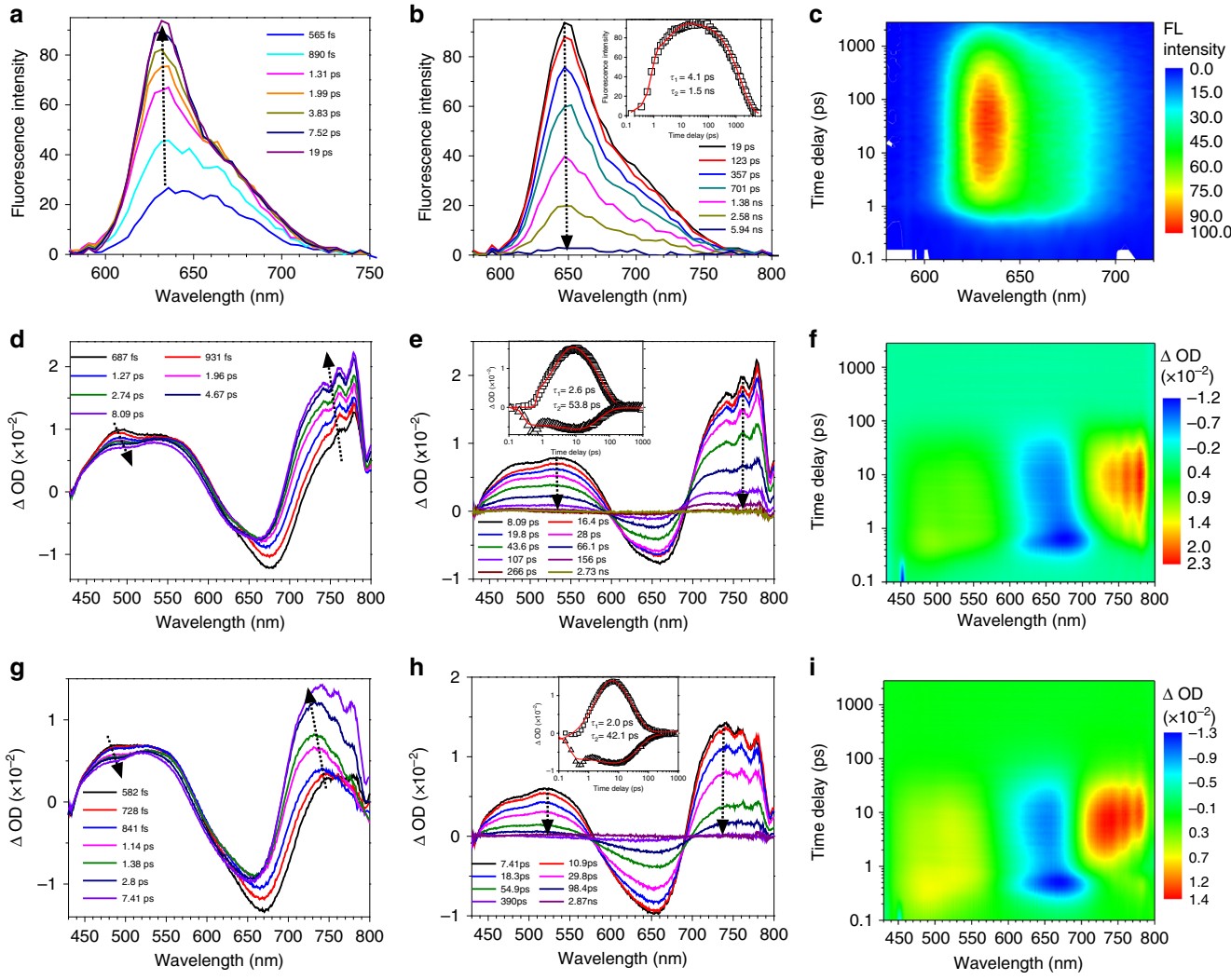

**Fig. 2** Excited-state dynamics of NDTA, 2TPE-NDTA, and 2TPE-2NDTA in solution observed by ultrafast time-resolved spectroscopy. **a**, **b** The fs-TRF spectra of NDTA obtained at time delay from 565 fs to 5.9 ns in THF after 400 nm excitation. Kinetic traces at 630 nm (black square) and solid lines (red) indicate the fitting trace to the experimental data points. **d**, **e** The fs-TA spectra of 2TPE-NDTA obtained at different time delays after 400 nm excitation. Kinetic traces at 730 nm (black square) and 630 nm (black triangle), solid lines (red) indicate the fitting trace to the experimental data points and the respective fit based on a global analysis with two exponential functions. **g**, **h** The fs-TA spectra of 2TPE-2NDTA at different time delays acquired after excitation at 400 nm. Kinetic traces at 726 nm (black square) and 629 nm (black triangle) and solid lines (red) indicate the fitting trace to the experimental data points and the respective fit based on a global analysis with two exponential functions. **c**, **f**, **i** Three-dimensional femtosecond transient emission and absorption spectra of NDYA2 (**c**), 2TPE-NDTA (**f**), and 2TPE-2NDTA (**i**). FL fluorescence, $\Delta OD = \log \frac{I_{100}}{I_T(t,\lambda)}$, $I_{100}$ the light level measured through the sample before excited states are created, $I_T$ transmitted (probe) light through the sample

varied contact time (CT) were implemented to assign the TPE peaks of 2TPE-NDTA. Then, the relaxation of the $^{13}C$ signals of the TPE moieties with time was recorded and plotted, affording the relaxation time of the TPE moieties to evaluate the intramolecular motion (Fig. 3b). Single TPE is a typical AIEgen with strong solid-state emission due to the suppression of intramolecular motion[36]. Indeed, a single TPE molecule shows a long relaxation time of around 5577 s in the solid state by $^{13}C$ CPMAS NMR measurement, demonstrating its hampered intramolecular motion. On the contrary, the relaxation time of the TPE moieties of 2TPE-NDTA is as short as 10.5 s, suggesting its much strengthened intramolecular motion. Similarly, 2TPE-2NDTA exhibits an even shorter relaxation time, indicating the more active intramolecular motion of its TPE moieties (Fig. 3c). It is worthy to note that the alkyl chains show an even shorter relaxation time than the TPE moieties (Supplementary Fig. 27), indicating their fluidity to benefit the intramolecular motion of

the TPE moieties. These data verify that even in the solid state, both 2TPE-NDTA and 2TPE-2NDTA still show strong intramolecular motion.

**Extending the molecular design to other system.** After demonstrating that the active excited-state intramolecular motion is chiefly responsible for the non-emissive nature of 2TPE-NDTA and 2TPE-2NDTA in aggregates, we hypothesize that the long alkyl chains of molecules play a decisive role in intermolecular spatial isolation to create some rooms to permit free intramolecular motion in the NP and solid state. To test the hypothesis, we synthesized 2TPE-NDTA and 2TPE-2NDTA with shorter alkyl side chains but failed due to the solubility issue. We thus chosen to synthesize their analogues to confirm this conjecture. As presented in Supplementary Fig. 2, two TPE-substituted perylenediimides (PDIs) called 2TPE-PDI-$C_6$ and 2TPE-PDI-$C_{16}$ with different alkyl side chain lengths were synthesized and

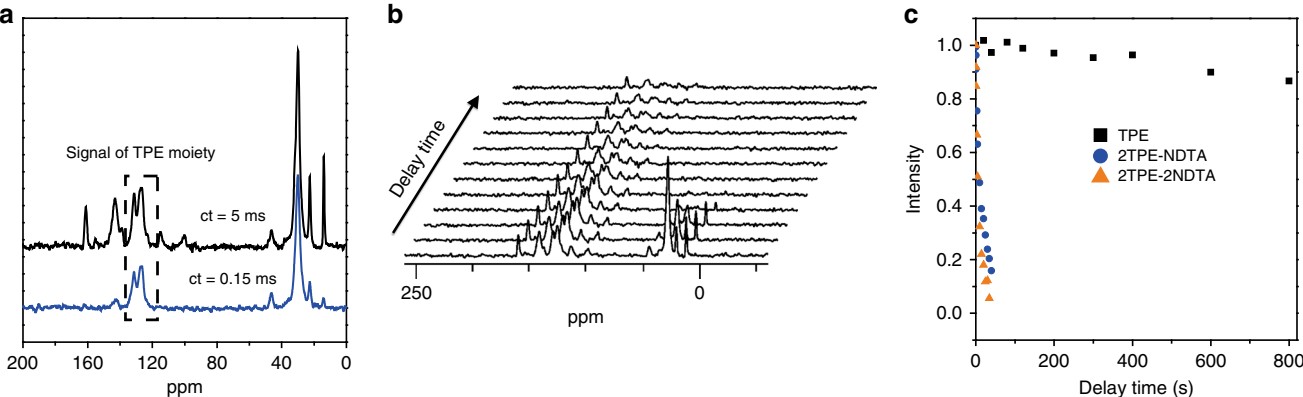

**Fig. 3** The SSNMR of TPE, 2TPE-NDTA, and 2TPE-2NDTA. **a** $^{13}$C CPMAS at 5 kHz with toss and varied contact time (CT) to assign the TPE peaks of 2TPE-NDTA. **b** $^{13}$C relaxation measurements of 2TPE-NDTA with a relaxation time of about 5577 s. **c** $^{13}$C relaxation measurements of 2TPE-NDTA (red) and 2TPE-2NDTA (green) with relaxation times of 10.5 and 7.1 s, respectively

characterized (Supplementary Fig. 14–19)[52]. As expected, by increasing the alkyl chain length form cyclohexyl in 2TPE-PDI-$C_6$ to 2-hexyloctyl in 2TPE-PDI-$C_{16}$, the QY in the film decrease dramatically from 17% to 6% due to the more active excited-state intramolecular motion. To confirm whether our hypothesis could be extended to other systems, reported semiconductors named TriPE-3PDIs with different side chains and strong red emission were synthesized (Supplementary Fig. 28) and their optical properties were compared[53]. The results indicate that TriPE-3PDIs with long alkyl chains of 2-hexyloctyl shows a much lower QY (6%) in the film state than its cousin with 2-ethylhexyl chains (30%). Therefore, the introduction of long alkyl chain into the backbone of molecular rotors is a general and effective strategy to promote the excited-state intramolecular motion and non-radiative relaxation of the excitons in aggregates. As the photophysical mechanisms of fluorescence and photothermy are opposite[54,55], the absorbed light energy when finally tends to lose as heat will result in reduced fluorescence.

**Photothermal conversion efficiency of the NPs**. The photothermal conversion of 2TPE-NDTA-doped and 2TPE-2NDTA-doped NPs in water was investigated as a practical biomedical application, operated in aqueous media. A semiconducting polymer NP (SPN) was reported as a high-performing photothermal agent and was used as a positive control[5,55] by the formulation of poly(cyclopentadithiophene-*alt*-benzothiadiazole) using DSPE-PEG$_{2000}$ as the matrix (Supplementary Fig. 29). Upon laser irradiation at 808 nm (0.8 W cm$^{-2}$), the temperatures of the aqueous solutions of all the NPs elevate with time and reach a maximum at 300 s (Fig. 4a). As shown in Fig. 4b, the plateau photothermal temperatures of 2TPE-2NDTA-doped NPs, 2TPE-NDTA-doped NPs, and SPNs are 81.4, 69.6, and 57.5 °C, respectively. According to the literature[56,57], the photothermal conversion efficiency (the efficiency of transducing incident absorbance to thermal energy) of SPNs is as high as 27.5%. Using the same calculation method, 2TPE-2NDTA-doped NPs and 2TPE-NDTA-doped NPs possess much higher photothermal conversion efficiencies of 54.9% and 43.0%, respectively, which are ultrahigh among currently available photothermal agents[56–58]. Such excellent photothermal behavior should be attributed to the strong light absorptivity and effective excited-state intramolecular motion within NPs that considerably harvest the absorbed light energy for heat production. This result manifests that active intramolecular motion within NPs is a highly efficient approach to induce photothermy.

**The mechanism comparison between AIE and iMIPT**. Based on the aforementioned results, we put forward a concept of intramolecular motion-induced photothermy (iMIPT) and compared its mechanism with that of AIE (Fig. 4c). Noteworthy, both AIE and iMIPT luminogens are rotor-rich, with similar excited-state molecular behaviors. In general, AIEgens show bright emission by suppression of active intramolecular motion in aggregates. On the contrary, iMIPT luminogens exhibit highly efficient photothermal conversion by active excited-state intramolecular motion in aggregates. No matter AIE or iMIPT, the excited-state intramolecular motion determines the photophysical property and the practical biomedical function and efficacy. Fundamentally, the iMIPT phenomenon deepens the understanding of the excited-state behavior of molecular rotors/vibrators and further confirms the crucial role of excited-state intramolecular motion in the photophysical properties of the luminogens. Furthermore, iMIPT makes it possible to utilize the non-radiative decay process of the luminogens to afford advanced photothermal nanoagents with excellent photothermal conversion efficiency, which provides opportunities for molecular rotors/vibrators to realize practical applications. It is also reasonable to anticipate that multifunctional materials with accurately controllable luminescence and/or photothermy are would be achieved by dint of regulating the activity of excited-state intramolecular motion.

**PA performance of the iMIPT NPs**. We next investigated the feasibility of conceptual iMIPT in advanced practical application. PA imaging of iMIPT NPs was studied as PA effect depends on photothermy. Before live animal study, the PA properties of various agents including NDTA-doped NPs, 2TPE-2NDTA-doped NPs, 2TPE-NDTA-doped NPs, SPNs, and MB were measured and compared. Their PA spectra at 680–980 nm are displayed in Fig. 5a. The maximum PA amplitudes of NDTA-doped NPs, 2TPE-NDTA-doped NPs, SPNs, and MB are all found at 680 nm, whereas 2TPE-2NDTA-doped NPs exhibit their PA maximum at 735 nm. The comparison of the PA signals of different PA agents was then performed upon 680 nm pulsed laser irradiation with a laser fluence of 17.5 mJ cm$^{-2}$ and a repetition rate of 10 Hz (Fig. 5b). At the same condition, noteworthy, 2TPE-NDTA-doped and 2TPE-2NDTA-doped NPs show about 5.9-fold and 7.6-fold higher PA intensities than NDTA-doped NPs. The significantly amplified PA signals are attributed to the TPE conjugation leading to iMIPT and bathochromic shift of absorption. Interestingly, 2TPE-2NDTA-doped NPs show the strongest PA signal among the tested agents in

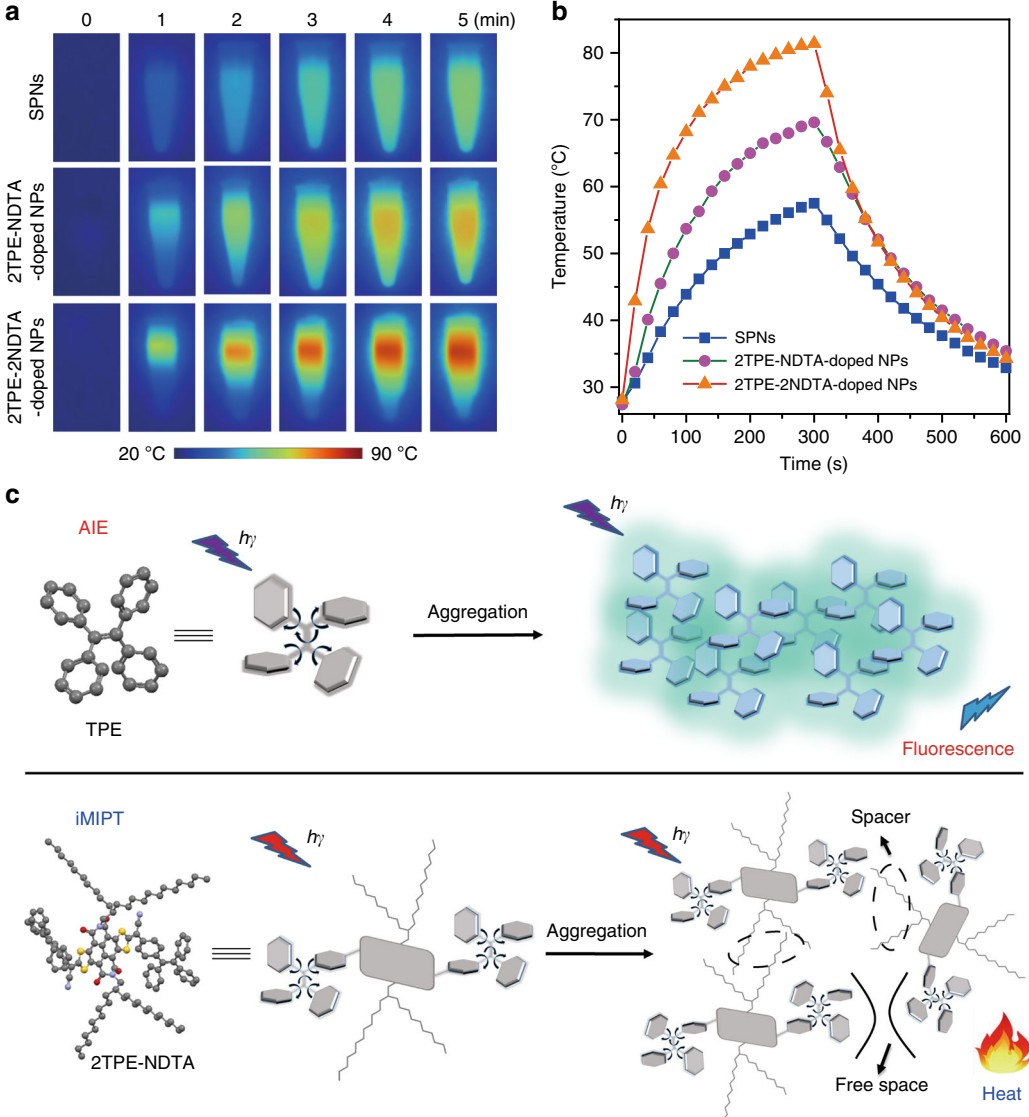

**Fig. 4** Photothermal conversion and working mechanism of iMIPT NPs. **a** IR thermal images of various NPs in aqueous solution (100 µM based on 2TPE-2NDTA, 2TPE-NDTA and the repeat unit of semiconducting polymer) upon exposure to 808 nm (0.8 W cm$^{-2}$) laser irradiation for different times. **b** The temperature changes of solutions of various NPs as a function of time. The solutions were irradiated with 808 nm laser (0.8 W cm$^{-2}$) for 300 s followed by naturally cooling for another 300 s. **c** The sketch map of working mechanisms of AIE and iMIPT

Fig. 5b with 680 nm pulsed laser even though 680 nm is not the optimized excitation wavelength. Furthermore, the PA intensities of both 2TPE-NDTA-doped and 2TPE-2NDTA-doped NPs are much higher than those of SPNs and MB. For example, the PA intensity of 2TPE-2NDTA-doped NPs is about 2.0-fold and 2.9-fold higher than that of SPNs and MB, respectively. It has been reported that SPNs are well-known, excellent PA contrast agents and exhibit an even better PA signal than single-walled carbon nanotubes[5]. Additionally, MB is also a widely used agent for PA imaging[4]. The comparison data reveal that iMIPT is a desirable platform for developing superior PA imaging probes and that the active excited-state intramolecular motion within NPs determines the biomedical function and effectiveness.

Figure 5c shows the PA intensities of 2TPE-NDTA-doped and 2TPE-2NDTA-doped NPs in phantoms irradiated by various numbers of laser pulses at 730 nm pulsed laser excitation with a laser fluence of 17.5 mJ cm$^{-2}$ and a repetition rate of 10 Hz. The result reveals that after exposure to $1.16 \times 10^8$ pulses with a total laser exposure time of 800 ms, there is negligible PA signal

decrease observed for both 2TPE-NDTA-doped and 2TPE-2NDTA-doped NPs, indicating their excellent photostability, which thus hold great potential for long-term and quantitative PA imaging.

The application of 2TPE-2NDTA-doped NPs in live-animal PA imaging was then studied and subcutaneous xenograft 4T1 breast tumour-bearing mouse model was employed. Before and after administration of 2TPE-2NDTA-doped NPs into the 4T1 tumour-bearing mice via the tail vein, in vivo PA imaging was conducted using a small-animal multi-spectral optoacoustic tomography (MOST) system with 730 nm pulsed laser excitation. Figure 5d depicts the time-dependent in vivo PA tumour imaging of 2TPE-2NDTA-doped NP-injected mice. Prior to NP administration (0 h), the tumours display weak PA signal at 730 nm because of the intrinsic background from oxyhemoglobin and deoxyhemoglobin[5,55]. After intravenous injection of 2TPE-2NDTA-doped NPs, intense PA signal can be clearly observed at the tumour site at 4 h post-injection. The PA intensity at 4 h is around 2.7-fold higher than that at 0 h (Fig. 5f). Such

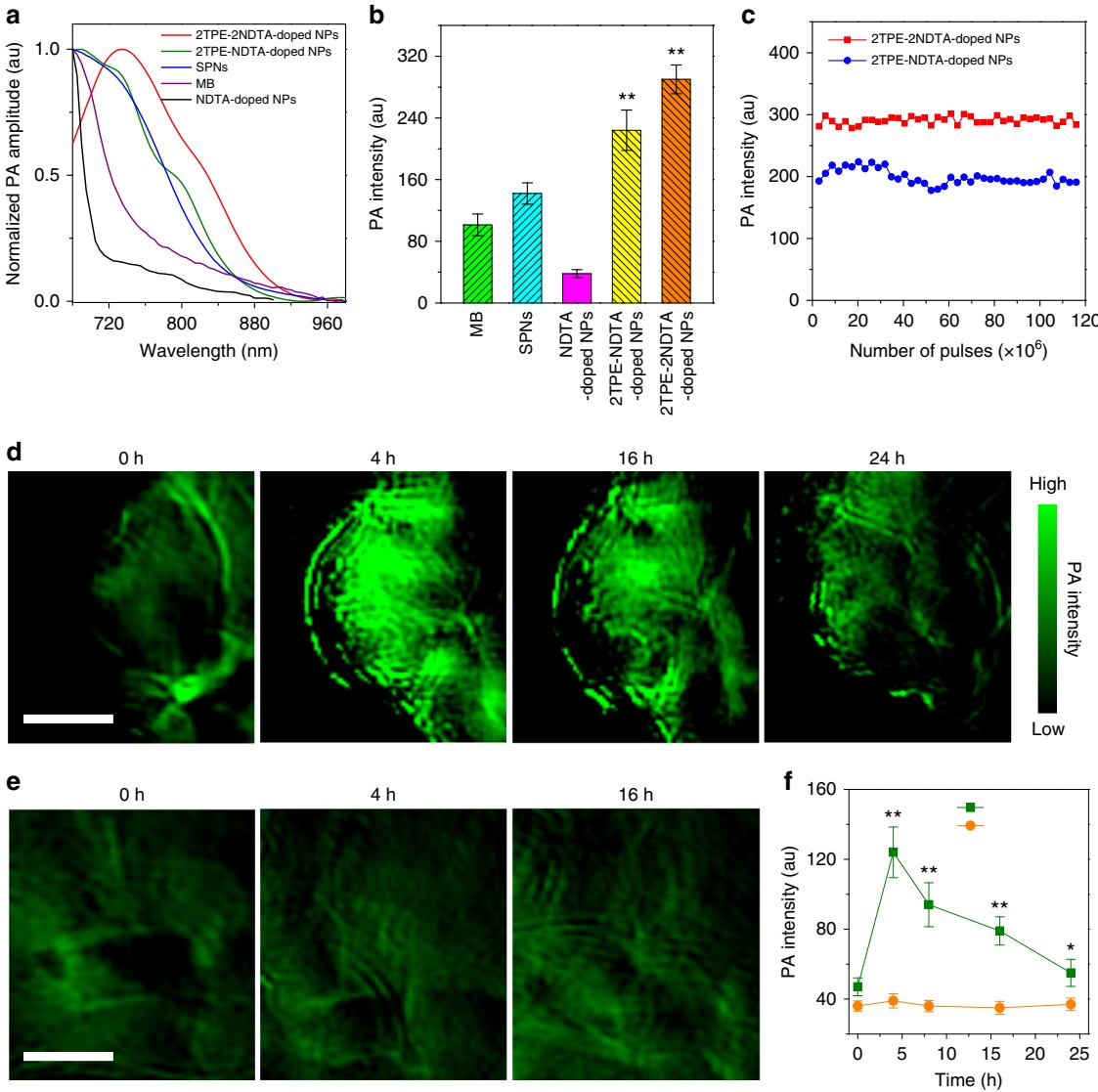

**Fig. 5** PA property and in vivo PA imaging with iMIPT NPs. **a** PA spectra of various NPs indicated. **b** PA signal (excited by 680 nm pulsed laser with a laser fluence of 17.5 mJ cm$^{-2}$ and a repetition rate of 10 Hz) comparison of different agents at the same concentration of 25 μM based on NDTA, 2TPE-NDTA, 2TPE-2NDTA, MB, and the repeat unit of semiconducting polymer. Error bars, mean ± s.d. (n = 3). **P < 0.01, in comparison between 2TPE-2NDTA-doped NPs or 2TPE-NDTA-doped NPs and other 3 agents (NDTA-doped NPs, SPNs, and MB) using one-way ANOVA. **c** Plot of PA intensities of 2TPE-2NDTA-doped and 2TPE-NDTA-doped NPs in phantoms against number of laser pulses at 730 nm. The total laser exposure time was 800 ms for 1.16 × 10$^8$ pulses. **d, e** Representative PA images of **d** tumours and **e** muscles from 2TPE-2NDTA-doped NP-administrated mice. Before (0 h) and after 2TPE-2NDTA-doped NPs (300 μM based on 2TPE-2NDTA) were intravenously injected into xenograft 4T1 tumour-bearing mice for designated time intervals, PA images were taken upon 730 nm pulsed laser irradiation with a laser fluence of 17.5 mJ cm$^{-2}$ and a repetition rate of 10 Hz. The ultrasound frequency is 5 MHz. Scale bars, 3 mm in both **d** and **e**. **f** PA intensities of tumour and muscle tissues as a function of time before (0 h) and after intravenous injection of 2TPE-2NDTA-doped NPs. Error bars, mean ± s.d. (n = 3 mice). **P < 0.01 and *P < 0.05, in comparison between tumour and muscle tissues using one-way ANOVA

high-contrast PA tumour imaging benefits from not only the excellent EPR effect of NPs but also the prominent PA effect of 2TPE-2NDTA in aggregated state. As a control, the muscle tissues from the same mice were also imaged under the same PA imaging condition before and after 2TPE-2NDTA-doped NP injection. The in vivo PA images over time and the corresponding signal dynamics from muscle tissues are shown in Fig. 5e, f. It is found that there is nearly no PA signal change in muscles among the tested time points (0, 4, 8, 16, and 24 h), and that the PA signal in muscle is significantly lower than that in tumour at each time point post-NP administration. The live animal results demonstrate that iMIPT NPs are capable of

serving as highly effective PA imaging probes for tumour diagnosis in vivo.

## Discussion

We have introduced a class of NIR-absorbing organic molecules. These molecules are non-fluorescent in the aggregate/solid state due to their iMIPT features. The iMIPT NPs with intrinsically active excited-state intramolecular motion show excellent photothermal conversion ability, which makes them desirable for building superior PA imaging probes for cancer diagnosis in live mice. By virtue of active excited-state intramolecular motion

within NPs, most absorbed energy can be harvested for heat generation, to permit advanced biomedical application, which is not attained by the currently existing materials such as molecular rotors and vibrators and molecular machines. This study thus demonstrates a strategy for developing advanced photothermal/ photoacoustic imaging nanoagents with a working principle different from the currently available ones. This study is also the first example of using molecular motion inside nanomaterials for practical application, which will inspire insights in both the fields of nanotechnology and molecular motion.

## Methods

**Femtosecond transient absorption (fs-TA) experiment.** Fs-TA measurement was done by using a femtosecond regenerative amplified Ti:sapphire laser system (1000 Hz, Maitai), and the amplifier was seeded with the 120 fs laser pulses from an oscillator laser system. The laser probe pulse was generated by employing ~5% of the amplified 800 nm laser pulses to produce a white-light continuum (430–750 nm) in a sapphire crystal, the probe beam then split into two parts before reaching the sample. One laser beam was focused on the sample while the other was focused on the reference spectrometer to monitor the fluctuations in the probe beam intensity. For the experiments in this work, THF solutions of the compounds were excited by a 400 nm pump beam (the second harmonic of the fundamental 800 nm from the regenerative amplifier). Solutions (1 mL) were studied in a 2 mm path-length cuvette, the absorbance throughout the data acquisition was 0.5 at 400 nm.

**Femtosecond time-resolved fluorescence (fs-TRF) experiment.** fs-TRF measurement was implemented on the same machine as fs-TA. An output laser pulse of 800 nm (200 mW) and a 400 nm laser pulse (10 mW) (second harmonic) were used as the gate pulse and pump laser, respectively. Upon excitation by the pump laser, the sample fluorescence was focused on the nonlinear crystal (BBO) mixing with the gate pulse to produce the sum frequency signal. Fluorescence spectra were then collected by changing the crystal angles and the spectra were probed by the air-cooled CCD. For the experiment in this work, THF solution of NDTA was excited by the 400 nm pump beam (the second harmonic of the fundamental 800 nm from the regenerative amplifier). Solutions (1 mL) were studied in a 2 mm path-length cuvette, the absorbance throughout the data acquisition was 0.5 at 400 nm.

**Solid-state NMR experiments.** NMR experiments were performed on a Varian Infinitplus-400 wide-bore (89 mm) NMR spectrometer at room temperature (25 °C) and frequencies of 399.72 and 100.52 MHz for $^{1}H$ and $^{13}C$, respectively. T3 probe with a rotor diameter of 4 mm was used, and samples with a volume of 52 μL were placed in a zirconia PENCIL rotor. The 90° pulse length was approximately 3 μs, corresponding to a radio frequency (RF) field strength of 83 kHz. The magic angle spinning (MAS) at 5 kHz was automatically controlled with a speed controller, and the total suppression of sidebands (TOSS) sequence was also used before the signal acquisition to suppress the spinning sidebands. The $^{13}C$ chemical shifts were referenced to external HMB (hexamethylbenzene, 17.3 ppm of $CH_3$). The ramped cross polarization (CP) was used for the $^{1}H$–$^{13}C$ polarization transfer, and the CP contact time was 0.1 and 1 ms, respectively.

**Preparation of NPs.** To a THF solution (1 mL), NDTA, 2TPE-NDTA, or 2TPE-2NDTA (1 mg) was dissolved. Then, DSPE-PEG$_{2000}$ (2 mg) was also added and dissolved in the THF solution. Afterward, the obtained THF solution was added into water (9 mL) accompanied with sonication using a microtip probe sonicator (XL2000, Misonix Incorporated, NY), followed by continuative sonication of the mixture for another 60 s. The THF in the mixture was evaporated by stirring in a fume hood for 12 h. The resulting NP suspension was purified by ultrafiltration (molecule weight cutoff 100 kDa) at 3000×$g$ for 0.5 h, which was subsequently filtered with a 0.2 μm syringe-driven filter.

**Cell culture.** Murine 4T1 breast cancer cells were purchased from American Type Culture Collection (ATCC). The cells were regularly checked for mycoplasma contamination. The 4T1 cancer cells were cultured in Dulbecco's Modified Eagle's Medium (DMEM) containing 10% fetal bovine serum (FBS) and 1% penicillin–streptomycin at 37 °C in a humidified environment containing 5% $CO_2$, respectively. Before experiments, the cells were pre-cultured until confluence was reached.

**Animals and tumour-bearing mouse model.** All animal studies were conducted under the guidelines set by Tianjin Committee of Use and Care of Laboratory Animals, and the overall project protocols were approved by the Animal Ethics Committee of Nankai University. Female BALB/c mice (6-week-old) were purchased from the Laboratory Animal Center of the Academy of Military Medical Sciences (Beijing, China). The xenograft 4T1 tumour-bearing mouse model was

used in the study. To set up the animal model, 30 μL of cell culture medium containing $1 \times 10^6$ murine 4T1 breast cancer cells were injected subcutaneously into the right axillary space of the BALB/c mouse. After about 10 days, mice with tumour volumes of about 80–120 mm$^3$ were used subsequently.

**Photostability study.** To assess the photostability, 100 μL of the NP aqueous solution (2TPE-NDTA-doped or 2TPE-2NDTA-doped NPs, 25 μM based on 2TPE-NDTA or 2TPE-2NDTA molecule) was added into the MOST phantoms, where optical and acoustic attenuation is minimal and the speed of sound is close to that of water (1480 and 1520 m/s at 21 and 34 °C, respectively). Every 30 s, the samples were exposed to $2.9 \times 10^6$ pulses within 20 ms at 730 nm pulsed laser excitation with a laser fluence of 17.5 mJ cm$^{-2}$ and a repetition rate of 10 Hz. Upon repeated 40 times, the total laser exposure time and total number of pulses were 800 ms and $1.16 \times 10^8$ pulses, respectively, during the process of evaluating the photostability of our NPs.

**In vivo PA imaging.** The 4T1 tumour-bearing mice were firstly anesthetized using 2% isoflurane in oxygen followed by intravenous injection of 2TPE-2NDTA-doped NPs (300 μM based on 2TPE-2NDTA) using a microsyringe via the tail vein ($n = 3$ mice). In vivo PA imaging was performed by a MOST imaging system (inVision 256-TF; iThera Medical, Germany). A wavelength-tunable (680–980 nm) optical parametric oscillator (OPO) pumped by an Nd:YAG laser provides excitation pulses with a duration of 7 ns at a repetition rate of 10 Hz. The light from the fiber covers an area of approximately 4 cm$^2$ with a maximum incident pulse energy of approximately 70 mJ at 730 nm (100 mJ, 70% fiber coupling efficiency). This generates an optical fluence of 17.5 mJ cm$^{-2}$, which is well within the safe exposures according to the American National Standard for Safe Use of Lasers. The signal detection is based on an ultrasonic cylindrically focused 270° transducer array (radius, 40 mm) with 256 evenly distributed detector elements achieving a center frequency of ultrasound at 5 MHz. In addition, the maximum in-plane resolution could reach approximately 150 μm with a section thickness of approximately 500 μm. In vivo PA images were acquired with 730 nm pulsed laser excitation before administration and at 4, 8, 16, and 24 h post-injection of 2TPE-2NDTA-doped NPs. After acquiring the imaging data, the images were reconstructed using the model-based algorithm supplied within the ViewMSOT software suite (V3.6, iThera Medical).

**Data analysis.** The investigators were not blinded to the group allocation. No data were excluded from the analysis. Quantitative data were expressed as mean ± standard deviation (s.d.). Statistical comparisons were made by one-way ANOVA. $P$ value < 0.05 was considered statistically significant. All statistical calculations were carried out with GraphPad Prism, including assumptions of tests used (GraphPad Software).

## Data availability

All relevant data that support the findings of this study are available from the corresponding authors upon reasonable request.

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

## Acknowledgements

This work was supported by the National Science Foundation of China (51622305, 51873092, and 21788102), the Innovation and Technology Commission (ITC-CNERC14SC01 and ITS/254/17), the National Basic Research Program of China (973 Program; 2015CB856503), the Research Grants Council of Hong Kong (16305015, 16308016, A-HKUST605/16, C2014-15G, and C6009-17G), and the Shenzhen Science and Technology Program (JCYJ20160229205601482 and JCYJ20160509170535223).

## Author contributions

Z.Z., D.D., and B.Z.T. conceived and designed the experiments. Z.Z and W.W. performed the synthesis. Z.Z and Y.X. did the PL measurement and analyzed the data. F.W. and P.S. performed the SS-NMR and analyzed the data. L.D. and D.L.P. performed the time-resolved spectra and analyzed the data. C.C. and X.Z. performed the PA imaging experiment. Y.C. performed the theoretical calculation. R.T.K.K., J.W.Y.L., X.H., and X.G. took part in the discussion and gave important suggestions. Z.Z., C.C., D.D., and B.Z.T. co-wrote the paper.

## Additional information

**Competing interests:** The authors declare no competing interests.

