## [Peer Review File · Nature Communications]

Reviewers' comments:

Reviewer #1 (Remarks to the Author):

Molecular imaging has been long considered as one of the most attractive advantages of optical imaging, by labeling the specific cellular components and tracking the biological processes of interest. However, it has also troubled the optical imaging community for years that high spatial resolution and large penetration depths of optical molecular imaging cannot be simultaneously achieved. Photoacoustic imaging (PAI) is proven a promising technology that may be a game changer in this battle, by acoustically probing the optical absorption contrast deep into the tissues.

The manuscript by Tang et al. has described a novel strategy of developing photothermal nanoagent for PAI. Although the concept of using photothermal nanoparticles in PAI has been demonstrated by other groups, this work is one step forward to improving this idea for biomedical research that needs penetration depth and molecular specificity. The manuscript was well written and easy to follow. The advantages of this work over previous publications were clearly stated. The potential biomedical impacts were justified. I applaud the authors' achievements.

The major contributions of the works can be put in two folds. (1) The concept of excited-state intramolecular motion is novel. By harnessing the non-radiative relaxation, the described work can be much more efficient for in vivo applications of photothermal therapy or photoacoustic imaging, with improved sensitivity and effectiveness. (2) The very careful study of the chemical and optical properties of the nanoagents is outstanding, which has provided a comprehensive understanding of the enhanced non-radiative relaxation process.

Although the chemical side of the work is excellent, I have a major concern about its significance to PAI. The reason is actually quite simple: the relatively low fluorescent quantum yield of most of the PAI contrast agents. I have tried to find the QY information of the three chemicals NDTA, 2TPE-NDTA and 2TPE-2NDTA, but maybe I have missed them in the manuscript. Nevertheless, the fluorescence quenching effect may be significant in 2TPE-NDTA and 2TPE-2NDTA, but the photothermal or photoacoustic enhancement might not be as significant. Let us just assume that the NDTA has a QY of 20%, and 2TPE-NDTA has a QY of 2%, then the radiative relaxation change is 10 folds. However, the non-radiative relaxation in this case increases from 80% to 98%, only 22% enhancement. Such a small improvement in thermal efficiency is actually not significant for PAI. This is a very common problem in development efficient PAI contrast agents: the change in fluorescent quantum yield most often does not result in substantial change in PA signal generation. A more practical way to improve the PA signals is to increase the absorption coefficient of the particles, shift the absorption peak towards the near-infrared window, and increase the Grueneisen coefficient. Therefore, my major concern about the present work is the limited improvement in the intended PAI applications may not be as significant as the authors have claimed.

I also have the following questions/suggestions for the authors' reference to strengthen the manuscript.

1. As mentioned above, it is critically important to clearly describe the fluorescence quantum yield of the three chemicals. The QY numbers will give the readers the necessary information to judge the potential improvement in the PA efficiency.
2. The photostability of the chemicals and NPs needs to be studied. Photobleaching is always a concern in quantitative PAI.
3. Line 59. The authors have done a good job describing the importance of non-radiative relaxation for PAI. However, optical absorption and Grueneisen coefficient are as important if not more important. Please add some discussion about the published efforts in improving these two.
4. Line 68, incomplete sentence 'are still looking forward to'.
5. Line 153, what is the definition of 'molar absorptivity'? If it is not a widely used term, please provide the molar extinction coefficient instead
6. Fig. 2f-i. what does the color bar represent? What does the negative value mean?
7. Line 181, 'NDYA2', typo?
8. Line 296, what is the definition of 'photothermal conversion efficiency'?
9. The PA experiments in Figure 5 have limited information for me to fully understand the imaging

process. For example, what is the optical fluence? what is the ultrasound frequency? what is the imaging reconstruction method?

10. Line 342, what is the reason for the PA absorption peak shift to 680 nm and 735 nm?

11. Fig. 5c. at 730 nm, hemoglobin still absorbs light and can provide contrast for PAI. Why is the image background so weak? should we see the signals from blood?

12. Fig. 5d, please also plot the signal dynamics from a non-tumor region as a control.

13. Fig. 5a, please also plot the PA spectrum for NDTA

14. Fig. 5b, please add the PA signals for NDTA.

15. Line 352, not sure MB is a 'star molecule' for PAI. maybe a 'widely used agent'?

16. The references lack representative PAI publications on the imaging systems.

Reviewer #2 (Remarks to the Author):

This article by Tang group describes "Highly efficient photothermal nanoagent achieved by harvesting energy via excited-state intramolecular motion within nanoparticles", Typically, author has described NIR-absorbing NDI bearing TPE molecules with the excited-state intramolecular motion-induced photothermy inside nanoparticles. Femtosecond time-resolved fluorescence and absorption spectroscopies and solid-state nuclear magnetic resonance studies used to investigate intramolecular motion. Due to presence of intramolecular motion realized in the solid state and within nanoparticles. Although this study is good extended study but not novel enough to publish in Journal like Nature Communication.

Following reasons justify rejection of this work to publish in Nature Communications:

1) There are many papers described similar phenomenon such as early work by Gärtner Phys. Rev. 122, 419-424 (1961) and recent work highlighted in ACS Nano 11, 7177-7188 (2017). J. Am. Chem. Soc. 139, 427 18115-18121 (2017).

2) Although reported molecular rotors (point 1 above) and vibrators generally show active intramolecular motion in solution state and not in aggregated and solid states, however, they give best performance rather than aggregated (particular) and solid state emission.

3) There are also many examples author has overlooked and not give the credit such as work by Jean-Pierre, S., who described very well these concepts

4) Just came across two paper when I type NDI and TPE I see it is about self-assembly in nanostructures author did not discussed them

5) Author should explain what it makes difference to use this work for PDT application as emission is higher in solid and aggregated states which is not giving by other dyes (emissive in solution and non-emissive in solid state)

6) The authors do not cite any well-known/cited prior works involving NDI tagged with TPE and other functional groups on to the core of NDI

7) Very recent work by Zhong and Yao (Angew.Chem.Int. Ed. 2018, 57,7820-7825) who shows similar strategy "Thermal-Responsive Phosphorescent Nanoamplifiers Assembled from Two Metallophosphors" i.e. emissive in aggregated states and there are many more examples in the literature.

8) So there is absolutely nothing novelty, either conceptually, or in terms of the types (and nature of) the examples described. The derivative nature of this paper (with addition of three new examples) makes this work seem incremental in the field. Thus, this work not novel enough to consider for publication in Nature Communication

There are two minor point author should also consider before submit to Sci. Rep.

1- In Figure 1c-f, the authors explained the fluorescence quenching of 2TPE-NDTA and 2TPE-2NDTA (in solution and solid state) by the non-radiative relaxation process is dominated for the exciton relaxation. Why the authors excluded other processes, in particular electron transfer process in both 2TPE-NDTA and 2TPE-2NDTA in solution?

2- Fs measurements (Fig. 2), I recommend to extend the scale of measurements to the NIR region. I think, the measurement in the NIR will give the authors chance to check if there is any generated transient species from transferring an electron/energy in both 2TPE-NDTA and 2TPE-2NDTA, and consequently give more clear picture about the mechanism of quenching process.

Therefore, I do not see this work has enough novelty to being of calibre for this journal. Having said/argue above points, I would recommend author should consider submitting this work in open access journal such as Scientific Reports.

Reviewer #3 (Remarks to the Author):

The manuscript by Tang and co-workers describes a creative new strategy to enable excited-state intramolecular motion for when dyes are in an aggregated state within NPs. Normally this motion is suppressed in the solid state or when aggregated however but appending long alkyl chains to the dye scaffold it creates a unique situation where the dye can still pi-stack while allowing the appended TPE moieties to rotate. The consequence of this is that the absorbed light can be dissipated via non-radiative relaxation pathways. The author leveraged this for both photothermal applications and photoacoustic imaging, both of which benefits from rapid non-radiative decay. The authors compared their dyes to other nanoparticles as well as small molecule dyes and in each instance their NPs exhibited superior performance. The authors performed a number of experiments including femtosecond time-resolved fluorescence, femtosecond transient absorption, and solid-state nuclear magnetic resonance to elucidate the mechanism of their design. Lastly the authors applied this to a photoacoustic imaging of tumors in live mice.

It is my opinion that the authors were able to come up with a very exciting new approach to harvest light to generate heat. Overall the authors put in tremendous effort in well planned out and executed experiments to support their findings. However, prior to recommending publication the authors must improve their writing. It is currently very poor quality and truly dampens the enthusiasm I have for this work. I will be willing to review this paper again after a rewrite has been submitted.

Responses to the Comments and Suggestions of Reviewer #1

Comments:

Molecular imaging has been long considered as one of the most attractive advantages of optical imaging, by labeling the specific cellular components and tracking the biological processes of interest. However, it has also troubled the optical imaging community for years that high spatial resolution and large penetration depths of optical molecular imaging cannot be simultaneously achieved. Photoacoustic imaging (PAI) is proven a promising technology that may be a game changer in this battle, by acoustically probing the optical absorption contrast deep into the tissues.

The manuscript by Tang et al. has described a novel strategy of developing photothermal nanoagent for PAI. Although the concept of using photothermal nanoparticles in PAI has been demonstrated by other groups, this work is one step forward to improving this idea for biomedical research that needs penetration depth and molecular specificity. The manuscript was well written and easy to follow. The advantages of this work over previous publications were clearly stated. The potential biomedical impacts were justified. I applaud the authors' achievements.

Response: We sincerely thank the appreciation and recognition of the reviewer on our work. We have provided a detailed point-to-point response to each question.

The major contributions of the works can be put in two folds. (1) The concept of excited-state intramolecular motion is novel. By harnessing the non-radiative relaxation, the described work can be much more efficient for in vivo applications of photothermal therapy or photoacoustic imaging, with improved sensitivity and effectiveness. (2) The very careful study of the chemical and optical properties of the nanoagents is outstanding, which has provided a comprehensive understanding of the enhanced non-radiative relaxation process. Although the chemical side of the work is excellent, I have a major concern about its significance to PAI. The reason is actually

quite simple: the relatively low fluorescent quantum yield of most of the PAI contrast agents. I have tried to find the QY information of the three chemicals NDTA, 2TPE-NDTA and 2TPE-2NDTA, but maybe I have missed them in the manuscript. Nevertheless, the fluorescence quenching effect may be significant in 2TPE-NDTA and 2TPE-2NDTA, but the photothermal or photoacoustic enhancement might not be as significant. Let us just assume that the NDTA has a QY of 20%, and 2TPE-NDTA has a QY of 2%, then the radiative relaxation change is 10 folds. However, the non-radiative relaxation in this case increases from 80% to 98%, only 22% enhancement. Such a small improvement in thermal efficiency is actually not significant for PAI. This is a very common problem in development efficient PAI contrast agents: the change in fluorescent quantum yield most often does not result in substantial change in PA signal generation. A more practical way to improve the PA signals is to increase the absorption coefficient of the particles, shift the absorption peak towards the near-infrared window, and increase the Grueneisen coefficient. Therefore, my major concern about the present work is the limited improvement in the intended PAI applications may not be as significant as the authors have claimed.

Response: We thank the reviewer so much for his/her extremely professional comments to improve our manuscript. We agree with the reviewer that it is a very common problem in development of NIR PAI contrast agents that the change in fluorescent quantum yield most often does not result in substantial change in PA signal generation. We think this could be mainly ascribed to the intrinsic nature of NIR-absorbing materials. According to the energy-gap law (*J. Phys. Chem.* **98**, 1289-7299 (1994)), the non-radiative decay of the luminescent materials is proportional to their band gap, which means smaller band gap affords more efficient non-radiative decay. Consequently, NIR-absorbing materials generally exhibit low fluorescent quantum yield and relatively efficient non-radiative decay. However, to further pursuit more efficient non-radiative decay, molecular design based on energy-gap law is limited because of smaller band gap usually needs stronger donor-acceptor structure, larger molecular π -conjugation as well as more complex

organic synthesis. In consideration of these, we put forward a new concept in our manuscript on alternative molecular design based on excited-state intramolecular motion, which can overcome this challenge elegantly since the introduction of molecular rotor is easy to manipulate in synthesis.

More importantly, different from the currently available systems with limited PA improvement, it is found that permitting excited-state intramolecular motion is a highly effective strategy to significantly enhance the PA signal generation of organic NIR materials. We apologize that we did not explain this issue in the previous manuscript, but in the revised version, we have clearly elaborated it by investigation of the PA property of NDTA according to the reviewer's comments. Firstly, we have supplemented the fluorescence quantum yields (QYs) of the NDTA, 2TPE-NDTA, and 2TPE-2NDTA in solution, nanoparticle (NP) and solid states (please refer to the response to Q1 for detailed information). Taking NPs for example, the QY values of 2TPE-NDTA-doped NPs and 2TPE-2NDTA-doped NPs are 0.4% and 0.3%, which are about 6.3-fold and 8.3-fold lower than that of NDTA-doped NPs (2.5%), respectively. On the other hand, the PA spectrum of NDTA-doped NPs were measured and the comparison of the PA signal among various PA agents including NDTA-doped NPs, 2TPE-NDTA-doped NPs, 2TPE-2NDTA-doped NPs, MB and SPNs was then performed with 680 nm pulsed laser (please refer to the responses to Q13 and Q14 for detailed information), and the result has been added as new Figures 5a and 5b in the revised manuscript. It is demonstrated that at the same experimental condition, NDTA-doped NPs show the weakest PA intensity due to the highest fluorescence in aqueous media. In contrast, 2TPE-NDTA-doped NPs and 2TPE-2NDTA-doped NPs show about 5.9-fold and 7.6-fold higher PA intensities than NDTA-doped NPs (Figure 5b in the revised version), which can be considered to be significant. The highly improved PA signals should be attributed to the tetraphenylethene (TPE) conjugation leading to excited-state intramolecular motion-induced photothermy (iMIPT) and bathochromic shift of absorption. As a consequence, as compared to the reported strategies for photothermal or PA enhancement that the reviewer raised, the effectiveness of iMIPT approach is more

significant. Moreover, 2TPE-NDTA-doped NPs and 2TPE-2NDTA-doped NPs also exhibit larger PA amplitude than MB and the well-known PAI high-performing SPNs (*Nat. Nanotechnol.* **9**, 233-239 (2014)), demonstrating iMIPT is an effective approach to advanced PA contrast agents.

In addition to the high PA enhancement efficacy, from the point of view of science, our manuscript demonstrates a new possibility to tune the excited state relaxation and deepen the understanding on excited molecular dynamics. Furthermore, it provides the chance to regulate the excited-state molecular motion to achieve the switch between radiative decay and non-radiative decay, to afford molecular materials with tunable and controllable fluorescent and photothermal properties. These thus justify the importance of this work as well.

I also have the following questions/suggestions for the authors' reference to strengthen the manuscript. Q1. As mentioned above, it is critically important to clearly describe the fluorescence quantum yield of the three chemicals. The QY numbers will give the readers the necessary information to judge the potential improvement in the PA efficiency.

Response: According to the reviewer's suggestion, we have supplemented the QYs of the three compounds in both solution, NP and solid states and added the data and corresponding discussion in the main text. The QY values are summarized in the following table.

	Quantum yield (QY)		
	NDTA	2TPE-NDTA	2TPE-2NDTA
Solution (soln)	17%	0.4%	0.4%
Solid	4%	1.0%	0.9%
NPs	2.5%	0.4%	0.3%

In the revised manuscript, we have added “As shown in Fig. 1d-f, NDTA emits brightly at around 630 nm in dilute THF solution, and its PL quantum yield (QY) was determined as $\Phi_{\text{soln, NDTA}} = 17\%$ using a calibrated integrating sphere. While in the aggregate state, both its NPs and film show much decreased PL intensity but largely red-shifted emission peaks at around 810 nm ($\Phi_{\text{NPs, NDTA}} = 2.5\%$, $\Phi_{\text{solid, NDTA}} = 4\%$), which suggests the formation of *J*-aggregation with aggregation-caused quenching (ACQ) characteristics (Supplementary Fig. S3)²². This is reasonable since the large π -conjugation and planar structure of NDTA is favorable for strong π - π stacking. However, after coupling with TPE, both the dilute THF solution and aggregates (NPs and thin film) of 2TPE-NDTA ($\Phi_{\text{soln, 2TPE-NDTA}} = 0.4\%$, $\Phi_{\text{NPs, 2TPE-NDTA}} = 0.4\%$, $\Phi_{\text{solid, 2TPE-NDTA}} = 1\%$) and 2TPE-2NDTA ($\Phi_{\text{soln, 2TPE-2NDTA}} = 0.4\%$, $\Phi_{\text{NPs, 2TPE-2NDTA}} = 0.3\%$, $\Phi_{\text{solid, 2TPE-2NDTA}} = 0.9\%$) emit almost no light (Fig. 1d-f), indicating that the non-radiative decay is dominated for the exciton relaxation of 2TPE-NDTA and 2TPE-2NDTA even in the solid state.” at line 13-21, page 8 and line 1-4, page 9 to describe and discuss the QY data.

Q2. The photostability of the chemicals and NPs needs to be studied. Photobleaching is always a concern in quantitative PAI.

Response: We strongly agree with the reviewer that the photobleaching is always a concern in quantitative PAI. Hence, we have investigated the PAI photostabilities of both 2TPE-NDTA-doped and 2TPE-2NDTA-doped NPs in aqueous media. In this experiment, 100 μL of the NP aqueous suspension (2TPE-NDTA-doped or 2TPE-2NDTA-doped NPs, 25 μM based on 2TPE-NDTA or 2TPE-2NDTA molecule) was added into the MOST (small-animal multi-spectral optoacoustic tomography)

phantoms, where optical and acoustic attenuation is minimal and the speed of sound is close to that of water (1480 and 1520 m/s at 21 and 34 °C, respectively). Every 30 s, the samples were exposed to 2.9×10^6 pulses within 20 ms at 730 nm pulsed laser excitation with a laser fluence of 17.5 mJ cm^{-2} and a repetition rate of 10 Hz. Upon repeated 40 times, the total laser exposure time and total number of pulses were 800 ms and 1.16×10^8 pulses, respectively, during the process of evaluating the PAI photostability of our NPs. According to the previous literature (*Nat. Nanotechnol.* **9**, 233-239 (2014), *Adv. Mater.* **27**, 5184-5190 (2015)), such experimental condition is high enough to assess the photostability of the PA imaging agents.

New Figure 5c in the revised manuscript (shown as follows) displays the PA intensities of 2TPE-NDTA-doped and 2TPE-2NDTA-doped NPs in MOST phantoms irradiated by various numbers of laser pulses. The result reveals that after exposure to 1.16×10^8 pulses with a total laser exposure time of 800 ms, there is negligible PA signal decrease observed for both 2TPE-NDTA-doped and 2TPE-2NDTA-doped NPs, indicating their excellent photostability, which thus hold great potential for long-term and quantitative PAI.

In the revised manuscript, we have added "Fig. 5c shows the PA intensities of 2TPE-NDTA-doped and 2TPE-2NDTA-doped NPs in phantoms irradiated by various numbers of laser pulses at 730 nm pulsed laser excitation with a laser fluence of 17.5 mJ cm^{-2} and a repetition rate of 10 Hz. The result reveals that after exposure to 1.16×10^8 pulses with a total laser exposure time of 800 ms, there is negligible PA signal decrease observed for both 2TPE-NDTA-doped and 2TPE-2NDTA-doped NPs,

indicating their excellent photostability, which thus hold great potential for long-term and quantitative PA imaging." at line 11-17, page 21.

In the revised Supplementary Information, we have also added "**Photostability study**. To assess the photostability, 100 μL of the NP aqueous solution (2TPE-NDTA-doped or 2TPE-2NDTA-doped NPs, 25 μM based on 2TPE-NDTA or 2TPE-2NDTA molecule) was added into the MOST (small-animal multi-spectral optoacoustic tomography) phantoms, where optical and acoustic attenuation is minimal and the speed of sound is close to that of water (1480 and 1520 m/s at 21 and 34 $^{\circ}\text{C}$, respectively). Every 30 s, the samples were exposed to 2.9×10^6 pulses within 20 ms at 730 nm pulsed laser excitation with a laser fluence of 17.5 mJ cm^{-2} and a repetition rate of 10 Hz. Upon repeated 40 times, the total laser exposure time and total number of pulses were 800 ms and 1.16×10^8 pulses, respectively, during the process of evaluating the photostability of our NPs." in the experimental section.

Q3. Line 59. The authors have done a good job describing the importance of non-radiative relaxation for PAI. However, optical absorption and Grüneisen coefficient are as important if not more important. Please add some discussion about the published efforts in improving these two.

Response: Thanks for this valuable suggestion of the reviewer. Under low-intensity irradiation, the PA response exhibits a linear dependence with respect to the incident light intensity as described by Eq. (1):

$$\text{PA} = \varepsilon_{\text{g}} C_{\text{g}} \Gamma I \Phi_{\text{nr}} \quad (1)$$

where ε_{g} is the molar extinction coefficient of the contrast agent at the incident wavelength, C_{g} is the ground-state concentration of dye molecules, Γ is the Grüneisen coefficient, I is the incident photon fluence, and Φ_{nr} is the quantum yield for nonradiative decay. The Grüneisen coefficient, Γ , is a constant that quantifies a medium's ability to conduct sound efficiently that is defined by Eq. (2):

$$\Gamma = V_{\text{s}}^2 \alpha / C_{\text{p}} \quad (2)$$

where V_s is the velocity of sound, α is the thermal expansion coefficient of the medium, and C_p is the specific heat of the medium at constant pressure.

According to the equations (1) and (2), the PA efficiency is correlated with several parameters such as the molar extinction coefficient, the Grüneisen coefficient, and the non-radiative decay efficiency as the reviewer mentioned. Among them, Grüneisen coefficient, Γ , is determined by several parameters such as the size of the nano PA agents and the matrix used in fabrication of nano PA agents (*Nanoscale* **7**, 337-343 (2015)). Moreover, the non-radiative decay efficiency and molar extinction coefficient is dominated by the dye loaded in the nano PA agents. Therefore, we strongly agree with the reviewer that optical absorption and Grüneisen coefficient are very important for improving the whole PA effect of the contrast agents. In the revised manuscript, to highlight the important role of optical absorption and Grüneisen coefficient in PAI, we have added some discussion about the published efforts in improving these two parameters, and the references “*ACS Nano* **12**, 1801-1810 (2018)”, “*J. Am. Chem. Soc.* **136**, 15853-15856 (2014)”, and “*Nanoscale* **7**, 337-343 (2015)” have been cited as ref. 17-19 in the revised version. The detailed changes in the manuscript to address this question are also shown as follows.

At line 3-14, page 3, “The PA effect of the materials that originated from heat generation is closely associated with several parameters of the contrast agents including the absorption coefficient, Grüneisen coefficient and the non-radiative decay efficiency according to the PA equation (see Supplementary Information)¹⁶. By either increasing the absorption coefficient or tuning Grüneisen coefficient, scientists have successfully improved the performance of the PA imaging agents¹⁷⁻¹⁹. For examples, Pu and Rochford et al. found that the increase of the ground state and excited absorption coefficient, respectively, could obviously enhance the PA intensity^{17,18}. Aoki and co-workers demonstrated that the thermal confinement in the nanoparticle and the large Grüneisen parameter of the material of nanosized PA contrast agents helped improve their PA performance significantly¹⁹. Besides, another important parameter is the non-radiative decay efficiency of the PA contrast agent, which determines how much absorbed light energy can be converted to heat¹⁶.”.

At line 7-10, page 9, "Additionally, the molar extinction coefficients of 2TPE-NDTA ($50600 \text{ mol}^{-1} \text{ L cm}^{-1}$) and 2TPE-2NDTA ($67800 \text{ mol}^{-1} \text{ L cm}^{-1}$) were comparable to that of NDTA ($67822 \text{ mol}^{-1} \text{ L cm}^{-1}$) (Supplementary Fig. S4), revealing their excellent light-harvesting ability, which could benefit to their PA signal according to the PA equation¹⁶".

At line 21-22, page 16 and line 1-2, page 17, "Such excellent photothermal behavior should be attributed to the strong light absorptivity and effective excited-state intramolecular motion within NPs that considerably harvest the absorbed light energy for heat production."

In the experimental section in revised Supplementary Information, page 6-7, "**PA equation.** Under low-intensity irradiation, the PA response exhibits a linear dependence with respect to the incident light intensity as described by Eq. (1)⁶:

$$PA = \varepsilon_g C_g \Gamma I \Phi_{nr} \quad (1)$$

where ε_g is the ground-state molar extinction coefficient of the contrast agent at the incident wavelength, C_g is the ground-state concentration of dye molecules, Γ is the Grüneisen coefficient, I is the incident photon fluence, and Φ_{nr} is the quantum yield for nonradiative decay. The Grüneisen coefficient, Γ , is a constant that quantifies a medium's ability to conduct sound efficiently that is defined by Eq. (2):

$$\Gamma = V_s^2 \alpha / C_p \quad (2)$$

where V_s is the velocity of sound, α is the thermal expansion coefficient of the medium, and C_p is the specific heat of the medium at constant pressure. "

Q4. Line 68, incomplete sentence 'are still looking forward to'.

Response: According to the reviewer's suggestion, this incomplete sentence has been revised as "while the latest breakthrough in molecular machine possibly bring us another giant leap in academics although exciting applications are still under investigation²⁸⁻³¹." at line 11-13, page 4 in the revised manuscript.

Q5. Line 153, what is the definition of 'molar absorptivity'? If it is not a widely used term, please provide the molar extinction coefficient instead.

Response: Molar absorptivity is a term with the same meaning as molar extinction coefficient. Both of them have been used in literatures (IUPAC, *Compendium of Chemical Terminology*, 2nd ed. (the "Gold Book") (1997)). However, we agree with the reviewer that "molar extinction coefficient" is a more widely used term. Therefore, we have changed all the "molar absorptivity" in the manuscript and Supplementary Information (Supplementary Figure 4) to "molar extinction coefficient".

Q6. Fig. 2f-i. what does the color bar represent? What does the negative value mean?

Response: The color bar in Figure 2c represents the fluorescence intensity, and the ones in Figure 2f and 2i represent the transient absorbance intensity. A laser flash photolysis spectrometer can monitor the transmitted (probe) light through the sample, at times during and shortly after an intense light pulse that converts a substantial amount of ground state molecules into molecules in the excited state. The change in the sample transmission allows the change in absorption to be calculated, typically in units of optical density, $\Delta OD = \log \frac{I_{100}}{I_T(t, \lambda)}$, I_{100} is the light level measured through the sample before excited states are created, I_T is the transmitted (probe) light through the sample. The negative value (band) of ΔOD means the ground bleaching in Figure 2d-i. In the revised manuscript, the labels of the color bars and the meaning of negative value have been clearly described in Figure 2 and its legend. New Figure 2 and the legend are also shown as follows.

Figure 2 | Excited-state dynamics of NDTA, 2TPE-NDTA and 2TPE-2NDTA in solution observed by ultrafast time-resolved spectroscopy. a,b, The fs-TRF spectra of NDTA obtained at time delay from 565 fs to 5.9 ns in THF after 400 nm excitation. Kinetic traces at 630 nm (black square) and solid lines (red) indicate the fitting trace to the experimental data points. d,e, The fs-TA spectra of 2TPE-NDTA obtained at different time delays after 400 nm excitation. Kinetic traces at 730 nm (black square) and 630 nm (black triangle), solid lines (red) indicate the fitting trace to the experimental data points and the respective fit based on a global analysis with two exponential functions. g,h, The fs-TA spectra of 2TPE-2NDTA at different time delays acquired after excitation at 400 nm. Kinetic traces at 726 nm (black square) and 629 nm (black triangle) and solid lines (red) indicate the fitting trace to the experimental data points and the respective fit based on a global analysis with two exponential functions. c,f,i, Three dimensional femtosecond transient emission and absorption spectra of NDTA (c), 2TPE-NDTA (f) and 2TPE-2NDTA (i). FL:

fluorescence. $\Delta OD = \log \frac{I_{100}}{I_T(t, \lambda)}$, I_{100} : the light level measured through the sample before excited states are created, I_T : transmitted (probe) light through the sample.

Q7. Line 181, 'NDYA2', typo?

Response: It is a typo, and "NDYA2" has been changed to "NDTA" in the revised manuscript.

Q8. Line 296, what is the definition of 'photothermal conversion efficiency'?

Response: According to the literature (*J. Phys. Chem. C* **111**, 3636-3641 (2007); *J. Am. Chem. Soc.* **138**, 9049-9052 (2016); *Nano Lett.* **11**, 2560-2566 (2011)), photothermal conversion efficiency can be defined as the efficiency of transducing incident absorbance to thermal energy. In the revised manuscript, we have added the definition of photothermal conversion efficiency "photothermal conversion efficiency (the efficiency of transducing incident absorbance to thermal energy)" at line 17-18, page 16. Furthermore, in the revised Supplementary Information, we have also added the calculation method of photothermal conversion efficiency in the experimental section, which is also shown as follows:

"Photothermal conversion efficiency calculation. Photothermal conversion efficiency (η) represents the efficiency of transducing incident absorbance to thermal energy, which could be calculated as follows, according to the literature³⁻⁵.

From an energy balance on a system, the total energy balance is:

$$\sum_i m_i C_{p,i} \frac{dT}{dt} = Q_{in, np} + Q_{in, surr} - Q_{out} \quad (1)$$

Where the i terms $m_i C_{p,i}$ are products of mass and heat capacity of system components. T is system temperature, and t is time.

$Q_{in,np}$ is the photothermal energy input from the agents, which can be described as:

$$Q_{in,np} = I(1 - 10^{-A_\lambda})\eta$$

Where I is the laser power in the photothermal experiment. A_λ is the absorbance at 808 nm.

$Q_{in,surr}$ is the heat input due to light absorption by the solvent and container, which can be described as:

$$Q_{in,surr} = Q_{Dis} = hS_{buff} \times (T_{Max} - T_{Surr})_{buffer}$$

Where hS_{buff} is the parameter relevant with container and solvent (h and S represent heat transfer coefficient and surface area of the container, respectively). $T_{max,buff}$ is the maximum steady-state temperature of solvent (without agents). T_{surr} is the ambient surrounding temperature.

Q_{out} is the heat lost to the surrounding, which can be described as:

$$Q_{out} = hS \times (T - T_{surr})$$

The hS and hS_{buff} can be determined by measuring the rate of temperature decrease after removing the light source.

At the maximum steady-state temperature, equation (1) equals to 0 and we can obtain:

$$Q_{in,np} + Q_{in,surr} = I(1 - 10^{-A_\lambda})\eta + Q_{dis} = Q_{out} = hS(T_{max} - T_{surr})$$

Where T_{max} is the maximum steady-state temperature of nanoparticles. As a consequence, η is determined by:

$$\eta = \frac{hS(T_{max} - T_{surr}) - Q_{dis}}{I(1 - 10^{-A_\lambda})} \quad "$$

Q9. The PA experiments in Figure 5 have limited information for me to fully understand the imaging process. For example, what is the optical fluence? what is the ultrasound frequency? what is the imaging reconstruction method?

Response: According to the reviewer's suggestion, in the revised manuscript and Supplementary Information, we have provided the detailed information on PA

experiments with the parameters including optical fluence, ultrasound frequency and imaging reconstruction method, which is also shown as follows.

In the experimental section of the revised version, we have added "**Photostability study**". To assess the photostability, 100 μL of the NP aqueous solution (2TPE-NDTA-doped or 2TPE-2NDTA-doped NPs, 25 μM based on 2TPE-NDTA or 2TPE-2NDTA molecule) was added into the MOST (small-animal multi-spectral optoacoustic tomography) phantoms, where optical and acoustic attenuation is minimal and the speed of sound is close to that of water (1480 and 1520 m/s at 21 and 34 $^{\circ}\text{C}$, respectively). Every 30 s, the samples were exposed to 2.9×10^6 pulses within 20 ms at 730 nm pulsed laser excitation with a laser fluence of 17.5 mJ cm^{-2} and a repetition rate of 10 Hz. Upon repeated 40 times, the total laser exposure time and total number of pulses were 800 ms and 1.16×10^8 pulses, respectively, during the process of evaluating the photostability of our NPs.

***In vivo* PA imaging.** The 4T1 tumour-bearing mice were firstly anesthetized using 2% isoflurane in oxygen followed by intravenous injection of 2TPE-2NDTA-doped NPs (300 μM based on 2TPE-2NDTA) using a microsyringe via the tail vein ($n = 3$ mice). *In vivo* PA imaging was performed by a MOST imaging system (inVision 256-TF; iTheraMedical, Germany). A wavelength-tunable (680-980 nm) optical parametric oscillator (OPO) pumped by an Nd:YAG laser provides excitation pulses with a duration of 7 ns at a repetition rate of 10 Hz. The light from the fiber covers an area of approximately 4 cm^2 with a maximum incident pulse energy of approximately 70 mJ at 730 nm (100 mJ, 70% fiber coupling efficiency). This generates an optical fluence of 17.5 mJ cm^{-2} , which is well within the safe exposures according to the American National Standard for Safe Use of Lasers. The signal detection is based on an ultrasonic cylindrically focused 270° transducer array (radius, 40 mm) with 256 evenly distributed detector elements achieving a center frequency of ultrasound at 5 MHz. In addition, the maximum in-plane resolution could reach approximately 150 μm with a section thickness of approximately 500 μm . *In vivo* PA images were acquired with 730 nm pulsed laser excitation before administration and at 4, 8, 16, 24 h post injection of 2TPE-2NDTA-doped NPs. After

acquiring the imaging data, the images were reconstructed using the model-based algorithm supplied within the ViewMSOT software suite (V3.6, iThera Medical).".

In the revised manuscript, we have added "The comparison of the PA signals of different PA agents was then performed upon 680 nm pulsed laser irradiation with a laser fluence of 17.5 mJ cm^{-2} and a repetition rate of 10 Hz (Fig. 5b)." at line 15-17, page 20, "Fig. 5c shows the PA intensities of 2TPE-NDTA-doped and 2TPE-2NDTA-doped NPs in phantoms irradiated by various number of laser pulses at 730 nm pulsed laser excitation with a laser fluence of 17.5 mJ cm^{-2} and a repetition rate of 10 Hz." at line 11-13, page 21, and "Before and after administration of 2TPE-2NDTA-doped NPs into the xenograft 4T1 tumour-bearing mice via the tail vein, *in vivo* PA imaging was conducted using a small-animal multi-spectral optoacoustic tomography (MOST) system with 730 nm pulsed laser excitation." at line 20-22, page 21.

Furthermore, some detailed PA experiments information has also been added in the legend of Figure 5, which includes "PA signal (excited by 680 nm pulsed laser with a laser fluence of 17.5 mJ cm^{-2} and a repetition rate of 10 Hz) comparison of different agents", "Plot of PA intensities of 2TPE-2NDTA-doped and 2TPE-NDTA-doped NPs in phantoms against number of laser pulses at 730 nm. The total laser exposure time was 800 ms for 1.16×10^8 pulses." and "Before (0 h) and after 2TPE-2NDTA-doped NPs ($300 \mu\text{M}$ based on 2TPE-2NDTA) were intravenously injected into xenograft 4T1 tumour-bearing mice for designated time intervals, PA images were taken upon 730 nm pulsed laser irradiation with a laser fluence of 17.5 mJ cm^{-2} and a repetition rate of 10 Hz. The ultrasound frequency is 5 MHz."

Q10. Line 342, what is the reason for the PA absorption peak shift to 680 nm and 735 nm?

Response: The PA absorption peak shift is very similar as the UV-vis-NIR absorption change. In the solution state (in THF), 2TPE-2NDTA shows almost the same absorption spectrum as that of 2TPE-NDTA because of its twisted molecular structure

in the solution state. However, in the aggregation state, the molecular planarity of 2TPE-2NDTA will be much improved and the intermolecular interaction will be enhanced, which lead to the red-shift of the absorption of 2TPE-2NDTA-doped NPs (Supplementary Figure 5). The PA absorption change of 2TPE-2NDTA NPs in aqueous media is consistent well with their UV-vis-NIR spectroscopy.

Supplementary Figure 5. a-c, UV-vis-NIR absorption spectra of NDTA (a), 2TPE-NDTA (b), and 2TPE-2NDTA (c) in THF solution, films and their doped organic NPs in water. Inset: photographs of the as-prepared NPs of NDTA, 2TPE-NDTA, and 2TPE-2NDTA in water.

Q11. Fig 5c. at 730 nm, hemoglobin still absorbs light and can provide contrast for PAI. Why is the image background so weak? should we see the signals from blood?

Response: We agree with the reviewer that at 730 nm, hemoglobin can absorb light and provide contrast for PAI. In the previous manuscript, the in vivo PA image background in Figure 5c is very weak because the PA signal from our NPs was separated from other intrinsic photoabsorbers such as oxyhemoglobin and deoxyhemoglobin, through linear spectral unmixing after image reconstruction using the ViewMSOT software suite (V3.6, iThera Medical). To address this issue clearly, the unmixed PA signal of NPs and the unmixed background of intrinsic photoabsorbers are presented in the same image with green and gray color, respectively, which are shown as follows.

In the revised manuscript, to avoid misunderstanding, we have deleted the unmixed PA signal images (previous Figure 5c) and added the PA images only after image reconstruction without linear spectral unmixing (new Figure 5d in revised version, which is also shown as follows).

Q12. Fig. 5d, please also plot the signal dynamics from a non-tumor region as a control.

Response: According to the reviewer's suggestion, we have provided the time-dependent PA images of mouse arm muscles from mice before (0 h) and after intravenous injection of 2TPE-2NDTA-doped NPs (new Figure 5e in the revised manuscript). The PA imaging condition is the same as that for tumour imaging.

Besides, the signal dynamics from muscle region is plotted and compared with that from tumour (new Figure 5f in the revised manuscript). The Figure 5e,f is shown as follows.

The result in Figure 5e,f reveals that there is nearly no PA signal change in muscles among the tested time points (0, 4, 8, 16, and 24 h), and that the PA signal in muscle is significantly lower than that in tumour at each time point post NP administration.

In the revised manuscript, we have also added "As a control, the muscle tissues from the same mice were also imaged under the same PA imaging condition before and after 2TPE-2NDTA-doped NP injection. The *in vivo* PA images over time and the corresponding signal dynamics from muscle tissues are shown in Fig. 5e,f. It is found that there is nearly no PA signal change in muscles among the tested time points (0, 4, 8, 16, and 24 h), and that the PA signal in muscle is significantly lower than that in tumour at each time point post NP administration." at line 7-13, page 22.

Q13. Fig. 5a, please also plot the PA spectrum for NDTA

Response: According to the reviewer's suggestion, we have measured and plotted the PA spectrum for NDTA-doped NPs in aqueous solution and added it in new Figure 5a in the revised manuscript, which is shown as follows.

The PA spectrum of NDTA-doped NPs is consistent well with their absorption spectrum (Supplementary Figure 5a), and the PA maximum of NDTA-doped NPs is located at 680 nm. In the revised manuscript, we have also added "Their PA spectra at 680-980 nm are displayed in Fig. 5a. The maximum PA amplitudes of NDTA-doped NPs, 2TPE-NDTA-doped NPs, SPNs and MB are all found at 680 nm" at line 13-15, page 20.

Q14. Fig. 5b, please add the PA signals for NDTA.

Response: We thank the reviewer very much for the valuable comment to improve our manuscript. According to the reviewer's suggestion, the PA signal of NDTA-doped NPs was measured and the comparison of the PA signal among various PA agents including NDTA-doped NPs, 2TPE-NDTA-doped NPs, 2TPE-2NDTA-doped NPs, MB and SPNs was performed with 680 nm pulsed laser (laser fluence of 17.5 mJ cm^{-2} and repetition rate of 10 Hz). The result has been added in new Figure 5b, which is shown as follows.

The data in Figure 5b indicate that at the same experimental condition, NDTA-doped NPs show the weakest PA intensity due to the highest fluorescence in aqueous media. In contrast, 2TPE-NDTA-doped NPs and 2TPE-2NDTA-doped NPs show about 5.9-fold and 7.6-fold higher PA intensities than NDTA-doped NPs, which can be considered to be significant. The highly improved PA signals should be attributed to the tetraphenylethene (TPE) conjugation leading to excited-state intramolecular motion-induced phototherapy (iMIPT) and bathochromic shift of absorption.

In the revised manuscript, we have also added the corresponding result and discussion "The comparison of the PA signals of different PA agents was then performed upon 680 nm pulsed laser irradiation with a laser fluence of 17.5 mJ cm^{-2} and a repetition rate of 10 Hz (Fig. 5b). At the same condition, noteworthy, 2TPE-NDTA-doped and 2TPE-2NDTA-doped NPs show about 5.9-fold and 7.6-fold higher PA intensities than NDTA-doped NPs. The significantly amplified PA signals are attributed to the TPE conjugation leading to iMIPT and bathochromic shift of absorption. Interestingly, 2TPE-2NDTA-doped NPs show the strongest PA signal among the tested agents in Fig. 5b with 680 nm pulsed laser even though 680 nm is not the optimized excitation wavelength. Furthermore, the PA intensities of both 2TPE-NDTA-doped and 2TPE-2NDTA-doped NPs are much higher than those of SPNs and MB. For example, the PA intensity of 2TPE-2NDTA-doped NPs is about 2.0-fold and 2.9-fold higher than that of SPNs and MB, respectively." at line 15-21, page 20 and line 1-5, page 21.

Q15. Line 352, not sure MB is a 'star molecule' for PAI. maybe a 'widely used agent'?

Response: According to the reviewer's suggestion, we have changed the previous description to "MB is also a widely used agent for PA imaging" at line 7-8, page 21 in the revised manuscript.

Q16. The references lack representative PAI publications on the imaging systems.

Response: We strongly agree with reviewer that the references in our manuscript lack representative PAI publications on the imaging systems, which should be cited. Therefore, in the revised manuscript, some representative PAI publications on the imaging system have been cited as ref. 6-15, which are shown as follows.

6. Razansky, D., Buehler, A. & Ntziachristos, V. Volumetric real-time multispectral optoacoustic tomography of biomarkers. *Nat. Protoc.* **6**, 1121-1129 (2011).
7. Razansky, D. *et al.* Multispectral opto-acoustic tomography of deep-seated fluorescent proteins in vivo. *Nat. Photon.* **3**, 412-417 (2009).
8. Yao, J. *et al.* High-speed label-free functional photoacoustic microscopy of mouse brain in action. *Nat. Methods* **12**, 407-410 (2015).
9. Li, L. *et al.* Single-impulse panoramic photoacoustic computed tomography of small-animal whole-body dynamics at high spatiotemporal resolution. *Nat. Biomed. Eng.* **1**, 0071 (2017).
10. Kim, C., Favazza, C. & Wang, L.V. In vivo photoacoustic tomography of chemicals: high-resolution functional and molecular optical imaging at new depths. *Chem. Rev.* **110**, 2756-2782 (2010).
11. Nie, L. & Chen, X. Structural and functional photoacoustic molecular tomography aided by emerging contrast agents. *Chem. Soc. Rev.* **43**, 7132-7170 (2014).
12. Zhang, E., Laufer, J. & Beard, P. Backward-mode multiwavelength photoacoustic scanner using a planar Fabry-Perot polymer film ultrasound sensor for high-resolution

- three-dimensional imaging of biological tissues. *Applied Optics* **47**, 561-577 (2008).
13. Bi, R. *et al.* Photoacoustic microscopy for evaluating combretastatin A4 phosphate induced vascular disruption in orthotopic glioma. *J. Biophotonics* **11**, e201700327 (2018).
 14. Wang, L. V. & Hu, S. Photoacoustic tomography: in vivo imaging from organelles to organs. *Science* **335**, 1458-1462 (2012).
 15. Zhang, H. F., Maslov, K., Stoica, G. & Wang, L. V. Functional photoacoustic microscopy for high-resolution and noninvasive in vivo imaging. *Nat. Biotechnol.* **24**, 848-851 (2006).

Responses to the Comments and Suggestions of Reviewer #2

Comments:

This article by Tang group describes “Highly efficient photothermal nanoagent achieved by harvesting energy via excited-state intramolecular motion within nanoparticles”, Typically, author has described NIR-absorbing NDI bearing TPE molecules with the excited-state intramolecular motion-induced phototherapy inside nanoparticles. Femtosecond time-resolved fluorescence and absorption spectroscopies and solid-state nuclear magnetic resonance studies used to investigate intramolecular motion. Due to presence of intramolecular motion realized in the solid state and within nanoparticles. Although this study is good extended study but not novel enough to publish in Journal like Nature Communication. Following reasons justify rejection of this work to publish in Nature Communications:

Q1. There are many papers described similar phenomenon such as early work by Gärtner Phys. Rev. 122, 419-424 (1961) and recent work highlighted in ACS Nano 11, 7177-7188 (2017). J. Am. Chem. Soc. 139, 427 18115-18121 (2017).

Response: We thank the reviewer for spending precious time to review our paper. For the references that the reviewer suggested, Gärtner’s work (*Phys. Rev.* **122**, 419-424 (1961)) used mathematic method to deduce the theory applicable to the photothermal effect of inorganic semiconductors such as germanium and silicon. Our previous work (*ACS Nano* **11**, 7177-7188 (2017)) reported the organic molecules for photothermal uses but was irrelevant with molecular motion. In another word, the PA effects of the agents in these two studies do not mainly benefit from the excited-state intramolecular motion. Differently, in our manuscript, we mainly demonstrate that permitting excited-state intramolecular motion is a highly effective strategy to significantly amplify the PA signal generation of organic NIR molecules. On the other hand, although Garcia-Garibay’s work achieved molecular rotor in crystal, whether their molecular rotor can work in amorphous state and random aligned aggregates is

unknown. More importantly, their molecules were not NIR-absorbing thus not related with PA and photothermal applications at all (*J. Am. Chem. Soc.* 139, 427, 18115-18121 (2017)). Therefore, the aforementioned studies mainly focused solely on either construction of molecular rotors or photothermal research of inorganic or traditional organic NIR molecules, and none of them was relevant with that using excited-state intramolecular motion within nanoparticles to harvest the absorbed light energy for heat production, which is the key novelty of our manuscript.

To help the reviewer better understand our manuscript, the novelty of our work is also summarized as follows: (1) we put forward a new concept of excited-state intramolecular motion induced photothermy (iMIPT), which is quite different from the molecular designs and working principles of currently reported photothermal materials, and iMIPT or similar concept has not been demonstrated in other studies so far; (2) the efficiency of traditional molecular photothermal system is limited by their aggregation state, and only H-aggregation with efficient non-radiative decay works well. Our iMIPT concept is not limited by this; by simply introducing molecular rotors, the exciton decays mainly through the non-radiative pathway, which can greatly improve the photothermal efficiency.; (3) the biological system is mostly hydrophilic environment, in which the organic molecules will form aggregates naturally. iMIPT thus enable the application of intramolecular motion within nanoparticles for bio-medical applications, which has not been achieved before; and (4) nanoparticles hold the key advantage of preferential accumulation in tumour tissues in vivo via the enhanced permeability and retention (EPR) effect (Langer, R. et al. *Science* **263**, 1600-1603 (1994), *ACS Nano* **3**, 16-20 (2009)). Our study hence not only provides a new strategy for developing advanced PA imaging nanoparticles through iMIPT, but also enables molecular motion in a nanoplatform to find a new way for practical application. Therefore, in consideration of these, our work is of large novelty and is distinguished from the suggested publications.

Q2. Although reported molecular rotors (point 1 above) and vibrators generally show active intramolecular motion in solution state and not in aggregated and solid states,

however, they give best performance rather than aggregated (particular) and solid state emission.

Response: We agree that current molecular rotors and vibrators work well in the solution state. However, for practical bio-medical applications of molecular rotors and vibrators, as the biological system is mostly hydrophilic environment, in which the organic molecules will form aggregates naturally. On the other hand, nanoparticles hold the key advantage of preferential accumulation in tumour tissues in vivo via the enhanced permeability and retention (EPR) effect (Langer, R. et al. *Science* **263**, 1600-1603 (1994), *ACS Nano* **3**, 16-20 (2009)). During the past 20-30 years, nanotechnology has greatly advanced the bio-medical field, particularly the diagnosis and treatment of cancers. Thereby, enabling molecular rotors and vibrators work in the nanoplatform decidedly boost the bio-medical applications of molecular rotors and vibrators as well as find a new outlet for the utilization of molecular motion. However, how to realize molecular rotors/vibrators with effective excited-state intramolecular motion in the aggregate state within nanoparticles is greatly challenging. Unfortunately, as the reviewer mentioned, the reported molecular rotors and vibrators generally show active intramolecular motion in solution, which is significantly suppressed in the aggregate and solid states. Thus, we are motivated to develop a new approach to alternative nanoparticles with highly boosted phototherapy by virtue of internally efficient excited-state intramolecular motion.

In our manuscript, to achieve effective live-animal PA tumour imaging application, the molecules (2TPE-NDTA and 2TPE-2NDTA) are rationally designed to realize NIR absorption and permit iMIPT within nanoparticles: (1) tetraphenylethylenen (TPE) is incorporated as it can undergo active excited-state intramolecular motion; (2) the acceptors are naphthalene diimide-fused 2-(1,3-dithiol-2-ylidene)acetonitriles (NDTA and 2NDTA) with long alkyl chains. The large π -conjugation and strong electron-withdrawing ability of the acceptors when combining with that of TPE will contribute long wavelength absorption, high molar absorptivity and strong twisted intramolecular charge transfer (TICT) effect to

the resulting molecules; and (3) the long alkyl chains enable the intermolecular spatial isolation of the molecules in the aggregate state to produce some necessary rooms to promote free intramolecular motion within nanoparticles. In the revised manuscript, we have added some corresponding discussions in the introduction part.

Q3. There are also many examples author has overlooked and not give the credit such as work by Jean-Pierre, S., who described very well these concepts.

Response: Prof. Jean-Pierre Sauvage demonstrated the first high-yielding template-directed synthesis of a [2]catenane, which uncovered the path that would ultimately lead others to the development of the field of mechanically interlocked molecules (MIMs) and ultimately to the discovery of molecular machines. In his recent Nobel lectures paper “From Chemical Topology to Molecular Machines (Nobel Lecture)”, he has discussed the importance of catenane and the relationship of catenane with molecular machine. Therefore, the concepts raised by Prof. Sauvage are different from that proposed by our work (please see the response to Q1 for the novelty of our work).

Actually, in the previous manuscript, we have cited Prof. Jean-Pierre Sauvage’s Nobel lecture paper (*Angew. Chem. Int. Ed.* 56, 11080-11093 (2017)) as ref. 18. Because we have added several references in the revised manuscript according to the reviewers' suggestions, the initial order of the reference is changed, the reviewer can see Prof. Jean-Pierre, Sauvage’s work in reference 31 in the revised version. Furthermore, the reference "*Acc. Chem. Res.* **34**, 477-487 (2001)" by Prof. Jean-Pierre Sauvage has also been cited as ref. 30 in the revised manuscript to support the description of molecular machines.

30. Collin, J.-P., Dietrich-Buchecker, C., Gaviña, P., Jimenez-Molero, M. C. & Sauvage, J.-P. Shuttles and muscles: linear molecular machines based on transition metals. *Acc. Chem. Res.* **34**, 477-487 (2001).

31. Sauvage, J. P. From chemical topology to molecular machines (Nobel lecture). *Angew. Chem. Int. Ed.* 56, 11080-11093 (2017).

Q4. Just came across two paper when I type NDI and TPE I see it is about self-assembly in nanostructures author did not discussed them

Response: We apologize very much for missing the works of Prof. Sheshanath V. Bhosale and co-workers about the TPE substituted NDI. They used the TPE substituted NDI to construct AIE materials. Differently, we introduce TPE to quench the emission in the aggregate and solid states. Therefore, the purposes of Prof. Bhosale's studies and our work are different. Since Prof. Bhosale's group reported for the first time on introduction of TPE into NDI moiety to construct AIE molecules, who also contribute importantly to this research area, in the revised manuscript, we have added the citations of Prof. Sheshanath V. Bhosale's studies, which have been cited as refs. 38, 40-43 in the revised version. Additionally, we have also added the corresponding discussion at line 4-6, page 9 in the revised manuscript.

Q5. Author should explain what it makes difference to use this work for PDT application as emission is higher in solid and aggregated states which is not giving by other dyes (emissive in solution and non-emissive in solid state).

Response: The reviewer possibly means PTT (photothermal therapy) rather than PDT (photodynamic therapy), as our work is irrelevant to PDT application. As compared to the dyes with emission in solution and non-emission in solid state, the difference and novelty of this work for PTT are summarized as follows:

(1) The concept of intramolecular motion induced photothermy (iMIPT) has been proposed and demonstrated, through which highly efficient photothermal materials could be afforded and showed better performance than the well developed methylene blue and polymer semiconductors nanoparticles (Figure 5 and Supplementary Figure 12). The main difference is that our work uses excited-state intramolecular motion within nanoparticles to harvest the absorbed

light energy for heat production, which is not achievable by currently reported systems.

- (2) For traditional molecular photothermal system, their photothermal efficiency depends on their aggregation state strongly. Only the one with H-aggregation exhibits efficient non-radiative decay and works well. However, since the aggregation formation was mostly random and difficult to control, the photothermal conversion efficiency of traditional photothermal agents is largely limited by their aggregation state. Our iMIPT concept is not limited by this; by simply introducing molecular rotors, the exciton decays mainly through the non-radiative pathway, which can greatly improve the photothermal efficiency.
- (3) Biological system is mostly hydrophilic environment, in which the organic molecules will form aggregates naturally. While in aggregates, intramolecular motion in most cases will be suppressed in a large extent, especially for NIR molecules with large molecular π -conjugation. iMIPT thus enable the application of intramolecular motion within nanoparticles for bio-medical applications, which has not been achieved before.
- (4) Nanoparticles hold the key advantage of preferential accumulation in tumour tissues in vivo via the enhanced permeability and retention (EPR) effect (Langer, R. et al. *Science* **263**, 1600-1603 (1994), *ACS Nano* **3**, 16-20 (2009)). Our work hence not only provides a new strategy for developing advanced PA imaging nanoparticles through iMIPT, but also enables molecular motion in a nanoplatform to find a new way for practical application.

Q6. The authors do not cite any well-known/cited prior works involving NDI tagged with TPE and other functional groups on to the core of NDI.

Response: We thank the reviewer very much for this suggestion. In the revised manuscript, we have cited some references by Prof. Sheshanath V. Bhosale and Prof. Daoben Zhu for their excellent works in functional NDIs, which are also shown as follows.

38. Al Kobaisi, M., Bhosale, S. V., Latham, K., Raynor, A. M. & Bhosale, S. V. Functional naphthalene diimides: synthesis, properties, and applications. *Chem. Rev.* **116**, 11685-11796 (2016).
39. Zhao, Z. et al. Naphthalenediimides fused with 2-(1,3-dithiol-2-ylidene)acetonitrile: strong electron-deficient building blocks for high-performance n-type polymeric semiconductors. *ACS Macro Lett.* **3**, 1174-1177 (2014).
40. Rananaware, A. La, D. D. & Bhosale, S. V. Solvophobic control aggregation-induced emission of tetraphenylethene-substituted naphthalene diimide via intramolecular charge transfer. *RSC Adv.* **5**, 63130 (2015).
41. Rananaware, A. La, D. D., Jackson, S. M. & Bhosale, S. V. Construction of a highly efficient near-IR solid emitter based on naphthalene diimide with AIEactive tetraphenylethene periphery. *RSC Adv.* **6**, 16250 (2016).
42. Bhosale, S. V., Jani, C. H. & Langford, S. J. Chemistry of naphthalene diimides. *Chem. Soc. Rev.* **37**, 331-342 (2008).
43. Bhosale, S. V., Bhosale, S. V. & Bhargava, S. K. Recent progress of core-substituted naphthalenediimides: highlights from 2010. *Org. Biomol. Chem.* **10**, 6455-6468 (2012).

Q7. Very recent work by Zhong and Yao (Angew. Chem. Int. Ed. 2018, 57, 7820–7825) who shows similar strategy “Thermal-Responsive Phosphorescent Nanoamplifiers Assembled from Two Metallophosphors” i.e. emissive in aggregated states and there are many more examples in the literature.

Response: We have carefully read Zhong and Yao’s paper (*Angew. Chem. Int. Ed.* 2018, 57, 7820–7825), the conclusion of their work is copied as follow:

“In conclusion, we have successfully constructed crystalline binary nanotubes from two iridium metallophosphors. The excellent light-harvesting and energy-transfer properties in these superstructures provide an efficient means for amplifying the acceptor emissions in crystalline states, which are otherwise

completely quenched upon aggregation. Moreover, the emissions of the obtained nanotubes show good sensitivity to thermal response. Color-tunable phosphorescence from green to red is realized in these structures by either varying the acceptor doping ratio or temperature. The thermochromic process is reversible with good photo- and structural stability and the emission color changes on isolated nanotubes are successfully recorded in situ. The thermally controlled exciton dynamics is considered responsible for the temperature modulated energy transfer and the luminescent thermochromism. This work provides an appealing alternative to overcome the ACQ issue of luminophores and achieve stimuli-responsiveness at the micro- and nanoscale, which would be of crucial significance for the applications of these materials in miniaturized photonic devices and solar energy harvesting."

According to their conclusion, Zhong and Yao's paper was mainly talking about how to overcome the ACQ problem and construction of thermal responsive phosphorescent system by a doping strategy, which mentioned nothing about molecular motion or photothermal effect. Differently, our work mainly focuses on using excited-state intramolecular motion within nanoparticles to harvest the absorbed light energy for heat production. As a result, the paper (*Angew. Chem. Int. Ed.* 2018, 57, 7820–7825) does not show similar strategy to our manuscript.

Q8. So there is absolutely nothing novelty, either conceptually, or in terms of the types (and nature of) the examples described. The derivative nature of this paper (with addition of three new examples) makes this work seem incremental in the field. Thus, this work not novel enough to consider for publication in Nature Communication.

Response: The novelty of our work has been summarized in the response to Q1. We hope this would help the reviewer understand our work.

There are two minor point author should also consider before submit to Sci. Rep. 1- In Figure 1c-f, the authors explained the fluorescence quenching of 2TPE-NDTA and

2TPE-2NDTA (in solution and solid state) by the non-radiative relaxation process is dominated for the exciton relaxation. Why the authors excluded other processes, in particular electron transfer process in both 2TPE-NDTA and 2TPE-2NDTA in solution?

Response: Since our group proposed the concept of aggregation-induced emission (AIE) in 2001, the underlying mechanism has been widely studied. In general, the mechanism of restriction of molecular motion (RIM) has been demonstrated and is widely accepted (*Adv. Mater.* **2014**, *26*, 5429–5479). RIM mechanism indicates that the twisted structure of AIE molecules such as TPE enable its active molecular motion in the solution state upon excitation, the active molecular motion promotes the rapid conformation change and thus the non-radiative decay to quench the emission. When aggregated, the molecular motion was suppressed and thus the non-radiative decay was inhibited, leading to the bright emission in the aggregate or solid state.

In this manuscript, our system obeys the RIM mechanism, which has been demonstrated by solvatochromism effect and low temperature experiment. In both non-polar solvent and low temperature environment (77k), the molecular motion is restricted and the fluorescence of 2TPE-NDTA and 2TPE-2NDTA will be lighted up (see the following Supplementary Figure 7 and 8).

Supplementary Figure 7. a,b, PL intensity of 2TPE-NDTA (a), and 2TPE-2NDTA (b) in THF solution at 298 K, solid at 298 K and THF solution at 77 K.

Supplementary Figure 8. a,b, PL intensities of **2TPE-NDTA (a)**, and **2TPE-2NDTA (b)** in different solvents.

For the emission quenched by electron transfer process (see the following images of Figs. 1 and 2 from the reference "*Analyst*, **2014**, *139*, 2641–2649"), it mainly occurs between the non-conjugated donor and acceptor system. It depends on the match of the HOMO and LUMO energy levels of the donor and acceptor moieties, respectively. The electron transfer process is not influenced by the temperature and solvent polarity, which means the emission quenched by PET process cannot be lighted up by changing temperature and solvent polarity. In consideration of these, we can exclude the electron transfer process.

Fig. 1 The mechanism of PET-based fluorescent probes.

Fig. 2 Frontier orbital theory for the PET effect.

2- Fs measurements (Fig. 2), i recommend to extend the scale of measurements to the NIR region. I think, the measurement in the NIR will give the authors chance to check if there is any generated transient species from transferring an electron/energy in both 2TPE-NDTA and 2TPE-2NDTA, and consequently give more clear picture about the mechanism of quenching process.

Response: As the Fs instruments in Hong Kong are not equipped with NIR, it is a pity that we cannot extend the scale of measurements to the NIR region. However, fortunately, please see the response to minor question 1, as we have reasonably excluded the possibility of electron transfer process, the NIR region Fs experiment is not necessary.

Responses to the Comments and Suggestions of Reviewer #3

Comments:

The manuscript by Tang and co-workers describes a creative new strategy to enable excited-state intramolecular motion for when dyes are in an aggregated state within NPs. Normally this motion is suppressed in the solid state or when aggregated however but appending long alkyl chains to the dye scaffold it creates a unique situation where the dye can still pi-stack while allowing the appended TPE moieties to rotate. The consequence of this is that the absorbed light can be dissipated via non-radiative relaxation pathways. The author leveraged this for both photothermal applications and photoacoustic imaging, both of which benefits from rapid non-radiative decay. The authors compared their dyes to other nanoparticles as well as small molecule dyes and in each instance their NPs exhibited superior performance. The authors performed a number of experiments including femtosecond time-resolved fluorescence, femtosecond transient absorption, and solid-state nuclear magnetic resonance to elucidate the mechanism of their design. Lastly the authors applied this to a photoacoustic imaging of tumors in live mice.

It is my opinion that the authors were able to come up with a very exciting new approach to harvest light to generate heat. Overall the authors put in tremendous effort in well planned out and executed experiments to support their findings. However, prior to recommending publication the authors must improve their writing. It is currently very poor quality and truly dampens the enthusiasm I have for this work. I will be willing to review this paper again after a rewrite has been submitted.

Response: We sincerely thank the reviewer for your careful reading and positive feedback to our work. According to the reviewer's suggestion, we have re-written the manuscript, which has also been revised by a native English speaker. In the revised manuscript, the changes have been highlighted in red for your reference.

REVIEWERS' COMMENTS:

Reviewer #1 (Remarks to the Author):

The authors have fully addressed my concerns in the first-round review, with new experimental data and detailed discussions. Looking from the photoacoustic imaging perspective, I think the manuscript is now ready for publication in Nature Communication.

Junjie Yao (Duke University)

Reviewer #2 (Remarks to the Author):

Author has done great work in this revised manuscript become wonderful article which will help to build strong platform harvesting energy via excited-state intramolecular motion. As English was bit my concern and author has done wonderful job to make much better flow in revised manuscript, which was required for journal like Nature. I also see author included many papers suggested by reviewer-2, really appreciated which was ignored by my comments.

Given shape and important of the work, I strongly recommend publication of this article in Nature Communication in its current form without any additional revision and corrections. I wish to see further development of field by author's.

Reviewer #3 (Remarks to the Author):

The authors have done a good job addressing the minor concerns brought up by all reviewers. Again, this is a new piece of work and has made advances in this important field. I am satisfied and support publication.